# Statistical structure of locomotion and its modulation by odors

Liangyu Tao[1], Siddhi Ozarkar[1], Jeffrey M Beck[2], Vikas Bhandawat[1,2,3]*

[1]Department of Biology, Duke University, Durham, United States; [2]Department of Neurobiology, Duke University, Durham, United States; [3]Duke Institute for Brain Sciences, Duke University, Durham, United States

**Abstract** Most behaviors such as making tea are not stereotypical but have an obvious structure. However, analytical methods to objectively extract structure from non-stereotyped behaviors are immature. In this study, we analyze the locomotion of fruit flies and show that this non-stereotyped behavior is well-described by a Hierarchical Hidden Markov Model (HHMM). HHMM shows that a fly's locomotion can be decomposed into a few locomotor features, and odors modulate locomotion by altering the time a fly spends performing different locomotor features. Importantly, although all flies in our dataset use the same set of locomotor features, individual flies vary considerably in how often they employ a given locomotor feature, and how this usage is modulated by odor. This variation is so large that the behavior of individual flies is best understood as being grouped into at least three to five distinct clusters, rather than variations around an average fly.
DOI: https://doi.org/10.7554/eLife.41235.001

## Introduction

There are many approaches to the study of neural underpinnings of behavior: One large body of work is rooted in the psychophysical literature where an animal is forced to choose between a few discrete behaviors (*Green and Swets, 1974*). This approach allows stimulus control and rigorous analysis of behavior based on an established framework (*Gold and Shadlen, 2000*), but sacrifices a full analysis of behavioral dynamics, leaving critical issues unexplored. Other studies have focused on behaviors that are reflexive (albeit with some flexibility) such as saccades (*Laurutis and Robinson, 1986*) and collision avoidance in insects (*Tammero and Dickinson, 2002*). Another large body of work has focused on the control processes involved in goal-directed behaviors such as reaching movements and has revealed many fundamental principles of motor control. Yet, another popular behavioral motif that has received much attention is behaviors that require meticulous sequencing (*Graybiel, 2008*). Finally, much work has been done to elucidate the workings of central pattern generators that underlie the rhythmic motor activity during walking and running (*Grillner, 1979*). Although many of these relatively stereotypical behavioral motifs are at play during most behaviors, they are not helpful in describing the structure underlying most everyday activities such as making a cup of coffee or a peanut-butter sandwich or walking to a car which consist of a sequence of actions, but neither the sequence nor each sub-action is stereotyped. These activities and their underlying sub-actions cannot be described either as sensorimotor reflexes or as behaviors that arise out of meticulous sequencing. An important example of such a behavior is an animal's locomotion. While tracks of a mouse or a fly exploring a chamber are not stereotypical there is an obvious structure to it.

Uncovering the structure within non-stereotyped behaviors such as locomotion requires sophisticated analytical tools. These tools can be applied to two complementary representations of an animal's behavior which can be described in the shape/posture space or in the world coordinate

*For correspondence:
vb37@duke.edu

**Competing interests:** The authors declare that no competing interests exist.

**eLife digest** Many behaviors that we perform everyday, including something as familiar as making a peanut-butter sandwich, consist of a sequence of recognizable acts. These acts may include, for example, holding a knife and opening a jar. Yet often neither the sequence nor the individual acts are always performed in the exact same way. For example, there are many ways to hold a knife and there are many ways to open a jar, meaning neither of these actions could be called "stereotyped".

A lack of stereotypy makes it difficult for a computer to automatically recognize the individual acts in a sequence. This same problem would apply to other common behaviors, such as walking around somewhere you have not visited before. While we easily recognize it when we see it, walking is not a stereotyped behavior. It consists of a series of movements that differ between individuals, and even in the same individual at different times. So how can someone automatically recognize the individual acts in a non-stereotyped behavior like walking?

To begin to find out, Tao et al. developed a mathematical model that can recognize the walking behavior of a fruit fly. Existing recordings of fruit flies walking were analyzed using a type of mathematical model called a Hierarchical Hidden Markov Model (often shortened to HHMM). Such models assume that there are hidden states that influence the behaviors we can see. For example, someone's chances of going skiing (an observable behavior) depend on whether or not it is winter (a hidden state).

The HHMM revealed that the seemingly random wanderings of a fly consist of ten types of movement. These include the "meander", the "stop-and-walk", as well as right turns and left turns. Exposing the flies to a pleasant odor – in this case, apple cider vinegar – altered how the flies walked by changing the time they spent performing each of the different types of movement. All flies in the dataset used the same ten movements, but in different proportions. This means that each fly showed an individual pattern of movement. In fact, the differences between flies are so great that Tao et al. argue that there is no such thing as an average walk for a fruit fly.

The model represents a complete description of how fruit flies walk. It thus provides clues to the processes that transform an animal's sensory experiences into behavior. But it also has potential clinical applications. Similar models for human behaviors could help reveal behaviors that are abnormal because of disease. Normal behaviors also show variability, and some diseases increase or decrease this variability. By making it easier to detect these changes, mathematical models could support earlier diagnosis of medical conditions.

DOI: https://doi.org/10.7554/eLife.41235.002

system. Recently much progress has been made in employing analytical tools that describe behaviors as a sequence of transformations in the shape/posture space using both supervised (*Branson et al., 2009*; *Kabra et al., 2013*), and unsupervised (*Berman et al., 2016*; *Berman et al., 2014*; *Wiltschko et al., 2015*; *Vogelstein et al., 2014*) algorithms. These studies provided remarkable insights by showing that much of an animal's behavioral repertoire can be described as transitions between few postures. While behavioral descriptions as transformations in posture space accurately classify the type of behavior (such as locomotion vs. grooming), because an animal's position in the world coordinate system is ignored, the structure within an animal's trajectory in world coordinates remains relatively unexplored.

Much of the work in extracting structure from an organism's trajectory is derived from the 'run and tumble model' which was originally employed in the context of bacterial chemotaxis (*Berg and Brown, 1972*). In this model, the organism is assumed to travel in relatively straight lines (runs) of exponentially distributed run lengths until they make sharp turns (tumble) to choose another direction at random. It is tempting to consider the motion of larger animals as roughly approximated by a run-and-tumble model, and many studies (explicitly and implicitly) employ a run-and-tumble framework with increasing sophistication as an analytical framework for locomotion (*Kim and Dickinson, 2017*; *Pierce-Shimomura et al., 1999*; *Schulze et al., 2015*). One obvious well-documented limitation of this framework is that animals do not turn at discrete times (*Stephens et al., 2010*; *Ohashi et al., 2014*; *Iino and Yoshida, 2009*; *Jung et al., 2015*; *Gomez-Marin and Louis, 2014*;

*Straub and Heisenberg, 1990*), and therefore, their locomotion cannot be described using a run-and-tumble framework. More generally, larger animals are likely to exert greater control over speed and direction of their locomotion and a better model is necessary to understand the resulting structure in their trajectory.

The lack of a model for locomotion makes it difficult to quantify the effect of stimuli on locomotion and is a critical missing piece in understanding the underlying sensorimotor transformations. For example, in many studies of odor modulation of locomotion, odors are primarily described as attractive or repulsive; this description is based on the end result, and does not consider the navigational maneuvers that underlie these end-results. Ignoring the underlying navigational maneuvers has led to a fundamental misunderstanding of odor modulation of locomotion. In a recent detailed analysis of a fly's locomotion, we demonstrated that it's navigational maneuvers in response to similarly attractive odors are quite distinct (*Jung et al., 2015*); the analysis used was based on an ad-hoc parametrization of locomotion, and not on a generative model of locomotion, making it difficult to determine whether our chosen parameter set was appropriate. A model of locomotion also makes it possible to compare how locomotion is affected by a given stimuli, and also how different individuals differ in their locomotion and in their response to stimuli.

In this study, we employ a hierarchical statistical model, Hierarchical Hidden Markov Model (HHMM) to describe the structure in the fly's locomotion (*Fine et al., 1998*). We show that fly locomotion is well-structured and an HHMM is an elegant representation of this structure. HHMM provides a simple and intuitive description of both a fly's locomotion and the effect of odors on the same. Surprisingly, different flies employ different strategies in their locomotion both before odor onset and in response to odors. Our data are, thus, inconsistent with the idea that the behavior of different flies represent variations around an 'average' fly. Rather, our data are most consistent with the idea that flies employ three to five different strategies, at a minimum, to explore a small circular arena and a similar number in their response to odors.

## Results

### Rationale for the choice of HHMM as the model and the model architecture

We model the locomotion of wild-type flies exploring a circular arena (*Jung et al., 2015*) whose center (odor-zone) consists of a fixed concentration of odor (*Figure 1A*). The arena and the experimental procedure was previously described (*Jung et al., 2015*). Briefly, locomotion of each of the 34 flies in our dataset was measured 3 min before an odor (apple cider vinegar or ACV) was turned on, and 3 min during the presence of ACV. Sample trajectories are shown in *Figure 1B*.

We first attempted to model the fly's locomotion using Hidden Markov Model (HMM) (*Gallagher et al., 2013*; *Isakov, 2016*). HMMs create discrete states based on a time series of observables such as position, speed or acceleration. The advantage of using HMMs in modeling locomotion is well described in earlier studies (*Gallagher et al., 2013*) (see Materials and methods).

In this study, we use observables that describe the change of position as a function of time, and hence our analysis will focus on behavioral states in the velocity space. Speed and angular speed are commonly used measures of velocity. But, because it is difficult to measure angular speed accurately at low speeds (*Gallagher et al., 2013*) (see Materials and methods section 2 for details), we fit the model to the component of speed parallel ($\widehat{v}_{||}$) and perpendicular ($\widehat{v}_{\perp}$) to its movement during the previous time point (*Figure 1C*, Materials and methods section 2). If the fly walks straight ahead, $\widehat{v}_{\perp}$ would be zero, therefore, $\widehat{v}_{||}$ and $\widehat{v}_{\perp}$ are closely related to speed and angular speed. We fit a time series of $\widehat{v}_{||}$ and $\widehat{v}_{\perp}$ to an HMM. The HMM architecture is shown in *Figure 1D*. We employed models with 24 to 50 states. HMM was only modestly successful because it was never able to classify >70% of the tracks into one of the states with >85% confidence.

The transition probability matrix for the HMM was sparse (*Figure 1—figure supplement 1*) suggesting that from each state there are transitions to only a handful of other states. One method to improve upon HMM performance is to cluster the states obtained by HMM according to the transition probability matrix. A common approach to clustering is to block-diagonalize the transition probability matrix (*Berman et al., 2016*). Tracks corresponding to the 10 states obtained by clustering are shown in *Figure 1—figure supplement 1*. Some of these states appear to describe recognizable

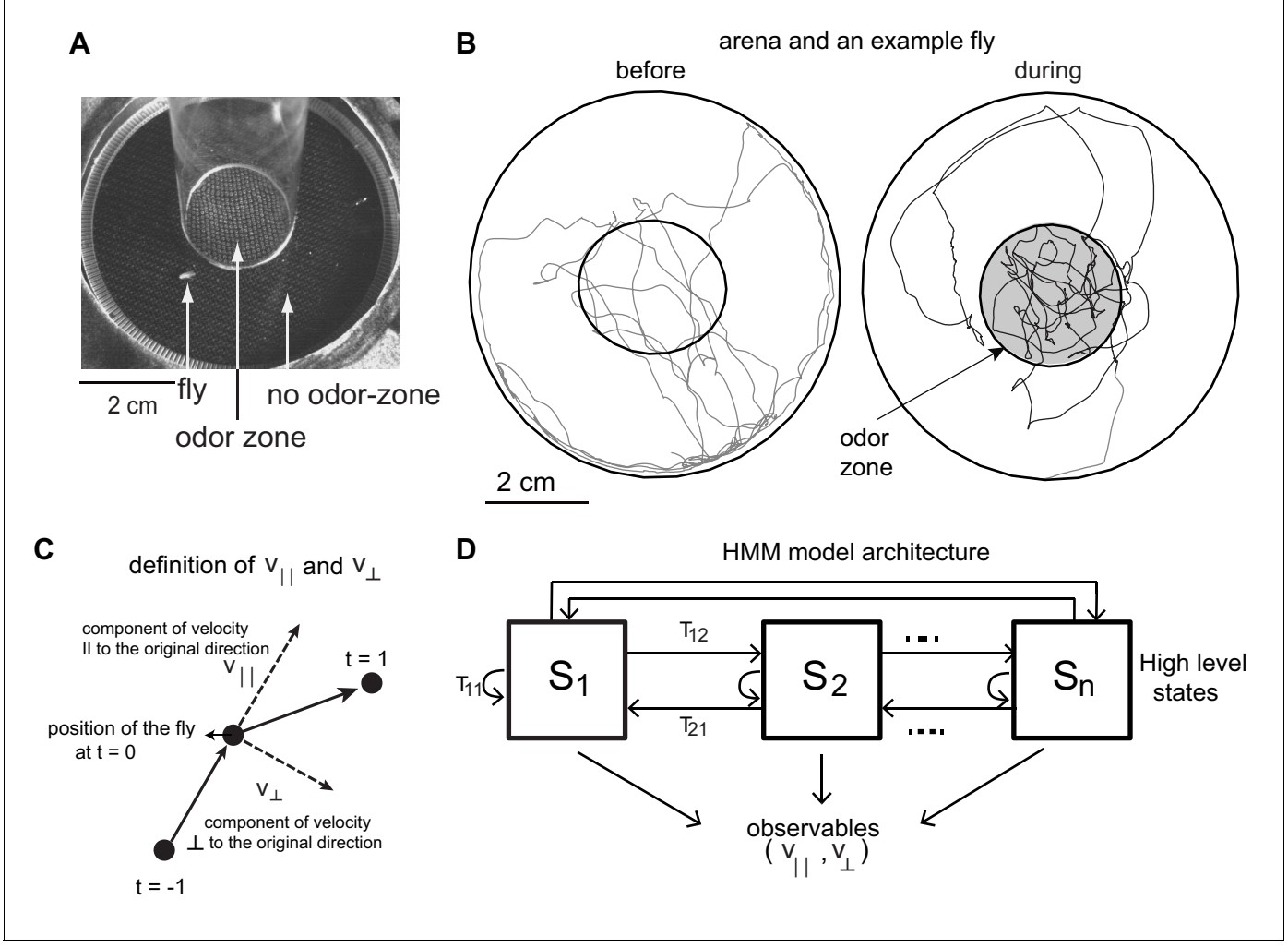

**Figure 1.** Experimental setup and HMM architecture. (**A**) Top view of the chamber. (**B**) Tracks of an example fly in a circular arena (3.2 cm in radius). The central region (1.2 cm radius) has no odor during the first 3 min (before period) and is odorized in the last 3 min (during odor). The odor zone is shaded. (**C**) The observables - $\hat{v}_{||}$ and $\hat{v}_{\perp}$ at time point t = 0 are schematized. $\hat{v}_{||}$ is the component of the fly's velocity along the velocity vector at the previous time point; $\hat{v}_{\perp}$ is the component perpendicular to the velocity vector at the previous time point. (**D**). HMM Model Architecture: A single layered model with *n* states each defined by a joint probablity distribution of the observables. The probability of transitioning from the $i^{th}$ state to the $j^{th}$ state is given by $T_{ij}$.

DOI: https://doi.org/10.7554/eLife.41235.003

The following figure supplement is available for figure 1:

**Figure supplement 1.** Block clustering of HMM states suggests a small number of locomotor features.

DOI: https://doi.org/10.7554/eLife.41235.004

features in the data such as left (state 10) or right turn (state 9). But efforts to block-diagonalize the transition probability matrix were only partially successful. The most obvious failure corresponds to the states with little movement. These states – describing the absence of movement – can occur in many different contexts, such as the fully stopped state or intermittent runs. When the same state is used in different contexts, an approximate block diagonalization of the transition probability matrix fails because the same state belongs to two blocks. In these cases, the different states that correspond to the absence of movement did not cluster, and appeared alone even after block-diagonalization (*Figure 1—figure supplement 1*). Thus, the existence of the same state in different contexts is one important reason for the modest success of HMM (*Marco et al., 2017*; *Wonjoon et al., 2010*; *Michele Weiland and Nelson, 2005*; *Nguyen et al., 2005*; *Murphy and Paskin, 2001*; *Chou, 2006*).

We employed a two-layered Hierarchical Hidden Markov model (HHMM) to model the data (**Figure 2A**). We reasoned that the low-level states (LL states) would be represented by Gaussian distributions on the observables, and the high-level states (HL states) would therefore be a mixture of Gaussians and would be able to model the experimental data better. Moreover, these HL states would have longer duration than the states discovered by HMM allowing it to more naturally model composite states.

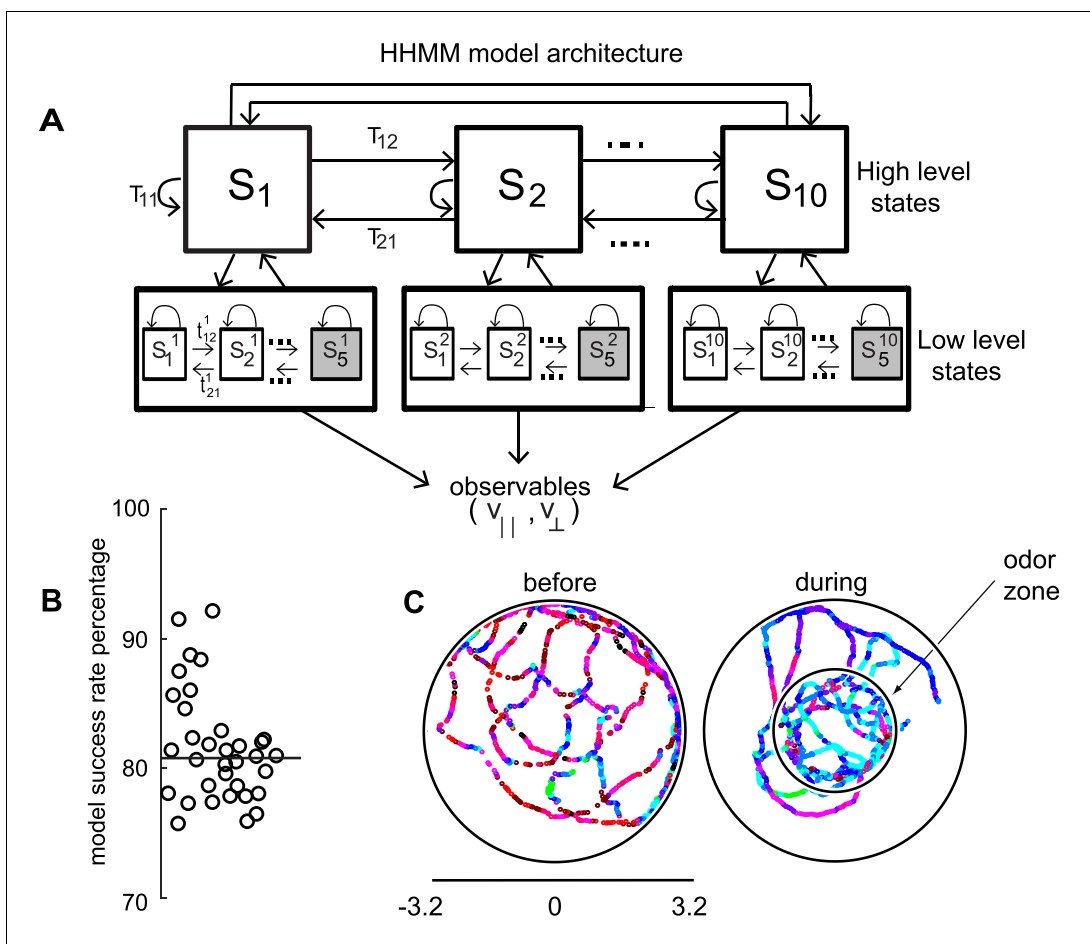

**Figure 2.** HHMM architecture. (**A**) Model architecture: The model consists of two layers; there are 10 high level states (HL states) each of which have five low level (LL) states. The probability of transitioning from the $i^{th}$ HL state to the $j^{th}$ HL state is given by $T_{ij}$. At the lower level, each HL state has its own transition probability matrix that describes transitions between its LL states. The shaded boxes represent the terminal states. (**B**) As a measure of the model's ability to fit the data for individual flies in our dataset - the percentage of timepoints for which the model had >85% confidence is plotted. Black line is the median. (**C**) HL state assignment for a single fly. The 10 high level states, each coded using a different color are overlaid on the tracks of a fly.
DOI: https://doi.org/10.7554/eLife.41235.005

The following figure supplements are available for figure 2:

**Figure supplement 1.** Longer duration of HHMM states allows it to discover structure in the data over longer times.
DOI: https://doi.org/10.7554/eLife.41235.006

**Figure supplement 2.** HL states have a longer duration than LL states because of multiple transitions among low-level states/transition between HL states.
DOI: https://doi.org/10.7554/eLife.41235.007

**Figure supplement 3.** Transition probability matrices.
DOI: https://doi.org/10.7554/eLife.41235.008

Indeed, in a Bayesian model comparison (see Materials and methods section 4), HHMMs outperformed HMMs with the same number of states. Since HMMs rarely used more than 35 states (even when models with higher number of states were fit), to perform model comparisons, we used models with less number of states than the particular HHMM we eventually employed. We compared a two-level HHMM with 6 HL states and 4 LL states to a single-level HMM with the same number of states ($4 \times 6 = 24$ states). We labeled the non-hierarchical model as the null model, and were able to reject the null model using Bayesian model comparison at $p < 0.0001$, implying that a hierarchical model is necessary. Another model comparison – a two-level model with 8 HL and 4 LL states compared to a single-level model with 32 states – also yielded similar results. The objectively better performance of an HHMM compared to an HMM suggests that a model that includes a hierarchical structure is more consistent with a fly's locomotion. It is important to note that, HHMMs are actually simpler than HMMs with the same number of states. This simplicity comes from the fact that any HHMM – which puts very specific constraints on the transition probability matrix – can be represented by an HMM but not vice-versa. HHMMs with the same number of states has far fewer parameters. Thus, for the two comparisons above, the HHMM has $6^2 + 6*4^2 + 6*4*5 = 252$, and $8^2 + 8*4^2 + 8*4*5 = 352$ parameters, and the HMM has $24^2 + 24*5 = 696$ and $32^2 + 32*5 = 1184$ parameters respectively; therefore, HHMM has fewer parameters. When a simpler model better characterizes the data, we can conclude that the additional structure contained in that model provides a more accurate characterization of the structure within the data.

The model we chose has 10 HL states (*Figure 2A*) and 5 LL states for each HL state. The model was fit to the entire dataset – both before and during the presence of the odors. The fitting process initializes by fitting each fly's tracks to its own HHMM and then clusters these 34 HHMMs – one for each fly – using a Gaussian mixture model, resulting in a smaller number of models. Remarkably, a single HHMM is an excellent fit for all the data suggesting that the behavior of wild-type flies is composed of similar components. The model was able to successfully assign an HL state (defined as >85% confidence) for >80% of the data points (*Figure 2B*, median 81%). This percentage was consistently high for all flies in our dataset (*Figure 2B*). In comparison, an HMM with 50 states can only classify 68% of the data with the same level of confidence. Tracks of a fly with each HL state labeled with a different color are shown in *Figure 2C*.

To make the difference between HHMM and HMM clearer, we compare the HHMM above to a HMM. As expected the time a fly spends in a HMM state is shorter than that in a HHMM state (*Figure 2—figure supplement 1A*). The longer time a fly spends in a HHMM state results from its hierarchical structure, and allows a HHMM to more accurately assign states, and is illustrated with two examples. First, consider a track that is assigned as a left turn by the HHMM, the HMM only classifies parts of the track as a left turn because of the inability of HMMs to consider longer duration trends in the observables (*Figure 2—figure supplement 1B1*). Short-term inhomogeneity in the data throws the HMM off; as soon as the $\widehat{v}_\perp$ decreases, the HMM exits its left turn state. Another example (*Figure 2—figure supplement 1B2*) shows that the HMM exits the stopped state as soon as there is a small movement. The net result is that the HHMM can classify all long stops into a single state while HMM needs four different stop states. HHMM also assigns more of the left turn as such (6% compared to only 2% by HMM).

The duration of the LL states of a HHMM is shorter than the duration of the HL state state (*Figure 2—figure supplement 2A*). Moreover, as the duration of HL states becomes longer, the mean number of transitions increase. The shorter duration of LL states compared to the HL states, and increased number of transitions between LL states within each HL state transition support the idea that there is structure at multiple timescales, and some of this structure is captured by the HHMM.

The limitations of HMM in describing phenomenon which have hierarchical and shared structure because of the short duration of its states is well documented (*Marco et al., 2017*; *Wonjoon et al., 2010*; *Michele Weiland and Nelson, 2005*; *Nguyen et al., 2005*; *Murphy and Paskin, 2001*; *Chou, 2006*). Therefore, an HHMM is an objectively a better model of fly walking data than an HMM.

## HL states of the HHMM model describe locomotor features

Both the organized transitions between HL states, and the narrow range of observables associated with each state shows that the fly's locomotion is structured: The transition probability matrix is sparse – a vast majority of transitions from each HL state were to 2-3 other HL states. When we

reordered the states (see Materials and methods section 5) from low-speed-high-turn-states to high-speed-low-turn states, we found that from any state the flies transitioned to the neighboring states with a high probability (*Figure 2—figure supplement 3*) suggesting a gradual transition from low-speed-high-turn states to high-speed-low-turn states. This gradual transition is not because flies cannot make large transitions due to biomechanical limitations because 47/81 1 possible transitions between HL states have a non-zero probability. Rather, transitions to states with similar kinematics show that under our experimental conditions – locomotion in a dark, small circular arena - flies locomote at similar $\widehat{v}_{\parallel}$ and $\widehat{v}_{\perp}$ for extended periods of time, and represent one way in which locomotion is organized.

More important to the organization is the narrow distribution of observables - $\widehat{v}_{\parallel}$ and $\widehat{v}_{\perp}$ - associated with each HL-state. The distribution of observables for a HL state is a composite of the distributions of its LL states (*Figure 3A*). Both the model (solid line in *Figure 3A1*) and a random sample of observables drawn from the time points assigned to a given LL state (gray markers) show that during each LL state within HL state 10, the observables are limited to a narrow range of values. In each LL state, $\widehat{v}_{\parallel}$ is large and $\widehat{v}_{\perp}$ is negative implying that in HL state 10, flies turn counter-clockwise at high speeds as observed in sample tracks corresponding to a single transition to HL state 10 (*Figure 3B*). The sample tracks also show that within each transition to a HL state there are multiple transitions between LL states, a signature of the hierarchical organization in our data. Fast, counter-clockwise turns represent a *locomotor feature* which describes a fly's locomotion in HL state 10. To better visualize this feature, we translated each track such that it began at the origin and rotated the tracks so

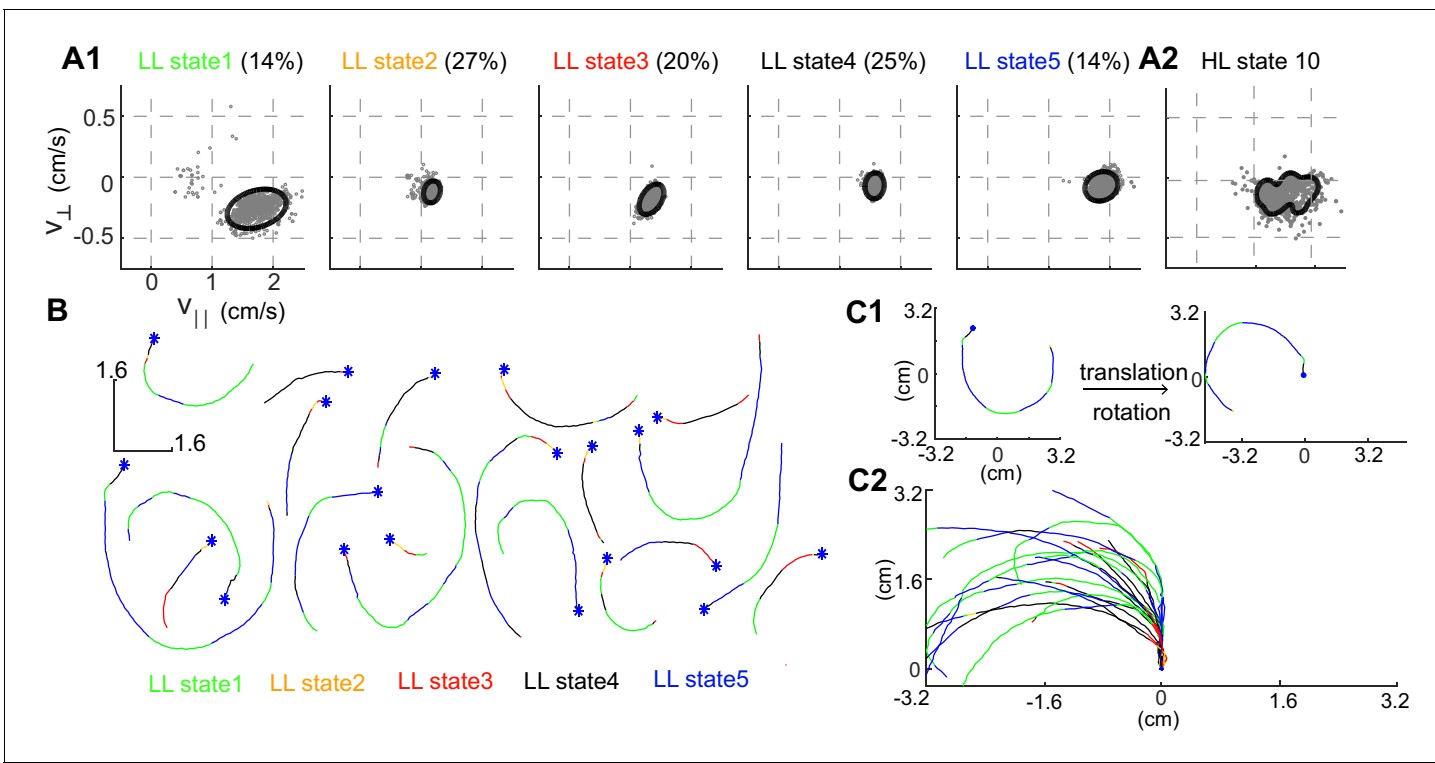

**Figure 3.** Structure of a HL state. (**A1**) 85% confidence bounds of the model (black ellipse) and a random sample of observables (gray dots) corresponding to data points assigned to the LL states underlying HL state 10. Percentage of time spent in a given LL state is also shown. (**A2**) Distribution of observables for the HL state. (**B**) Example tracks denoting a single transition to HL state 10 show that the fly is turning counterclockwise. LL states were color coded. (**C1**) Each track is rotated and translated for visualization. (**C2**) All 20 tracks in B were transformed as shown in C1. Transformation reveals that all HL state 10 trajectories represent left turns.

DOI: https://doi.org/10.7554/eLife.41235.009

The following figure supplement is available for figure 3:

**Figure supplement 1.** LL states corresponding to each HL statest: The same plot as *Figure 3A* for all the states.
DOI: https://doi.org/10.7554/eLife.41235.010

that the initial velocity vector pointed along the y-axis (*Figure 3C₁*, see Materials and methods). These transformations make it apparent that the locomotor feature for state 10 is turning left at high speeds (*Figure 3C₂*).

Rotated and translated (as in *Figure 3C*) tracks for each of the 10 HL states are shown in *Figure 4*. The distribution of the observables for each HL state is also plotted. HL state 1 represents very slow walking with frequent changes in direction. In state 2, flies are either completely stopped or they

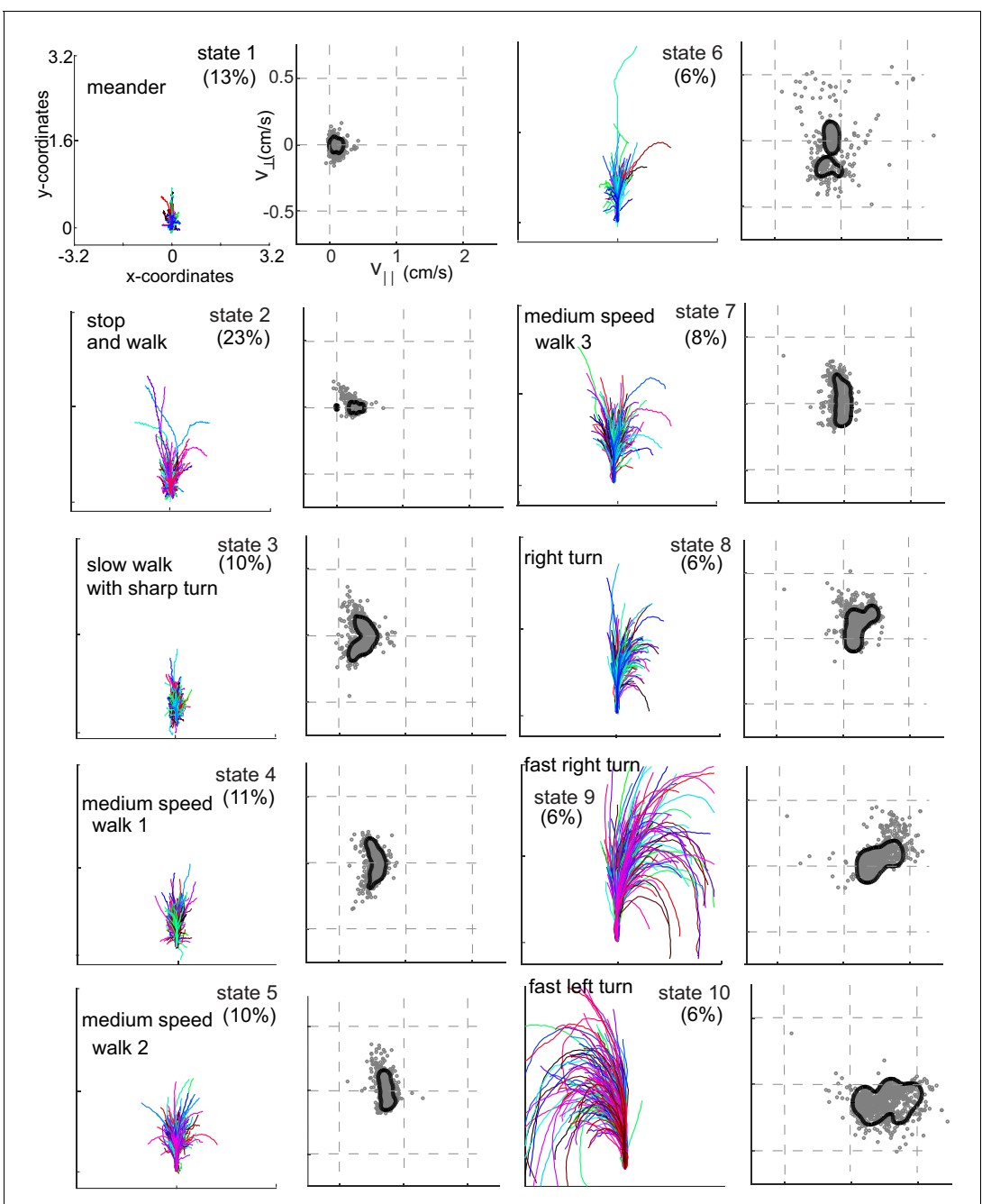

**Figure 4.** Each HL state describes a locomotor feature during which the fly uses a narrow region of the velocity space. Trajectories (left) and distribution of observables (right) corresponding to each of the 10 HL states are shown. Trajectories were transformed via translation and rotation to start at the origin, and the initial velocity vector pointed along the y-axis. Both model distribution (solid lines) and randomly selected 1000 empirical datapoints (gray dots) belonging to a given state are shown. The percentage of time a fly spends in a given HL state is shown.
DOI: https://doi.org/10.7554/eLife.41235.011

walk at a speed about twice the speed of the fly in state 1; state 2 represents stop and start locomotion. The subtle, but important differences between state 1 and state 2 show an instance in which the HHMM is successful at extracting an unexpected feature in the velocity profile in a fly's locomotion. During state 3, the fly is exhibiting a sharp turn that is reflected in the increase in $\widehat{v}_\perp$ with a concomitant decrease in $\widehat{v}_\parallel$. These three states together represent slow locomotion.

In states 4-7, flies are walking at a medium-speed. In contrast to the clear drop in $\widehat{v}_\parallel$ with increases in $\widehat{v}_\perp$ in state 3, $\widehat{v}_\parallel$ remains strikingly constant irrespective of $\widehat{v}_\perp$. These states are different from each other because $\widehat{v}_\parallel$ is slightly different.

States 8–10 are high-speed states; each of these states is also characterized by their turn direction. During states 8 and 9, the fly turns right; the fly's speed is higher during state 9 than during state 8. During state 10, the fly turns left. States 9 and 10 are mirror-symmetric versions of each other.

Flies spend 60% of time performing a locomotor feature for >300 ms, and >10% of their time performing a single locomotor feature for >3 s (*Figure 2—figure supplement 2*). Thus, flies spend extended time in the same state.

## Odors affect locomotion by altering the occupancy of HL states

In the absence of ACV, the state occupancy inside and outside the odor-zone are quite similar: The fly spends 30% of its time in state 2 and roughly equal time in all other HL states. Introducing ACV changes the fly's locomotor behavior both inside and outside the odor-zone, but with opposite effects on the HL state occupancy in the two zones. Inside the odor-zone, in the presence of ACV, the fly spends more time in HL states 1 and 3 at the expense of time spent in HL states 7–10 (*Figure 5A*). These changes from high-speed states to low-speed states suggest that in the presence of ACV the fly is performing a local search, presumably to find food. Outside the odor-zone (*Figure 5B*), the fly spends more time in the high-speed states (HL states 8–10), with a decrease in the occupancy of HL state 2 (which includes stopping). Decreased stopping and increased high-speed walking with turning is likely to represent a different search strategy, wherein the fly might be attempting to re-find the odor it has recently lost. We also investigated whether there were changes in the LL state composition of the HL states and found no changes (*Figure 5—figure supplement 1*). Overall, these results showed that odors affect locomotion not by creating new locomotor features, but by altering the frequency with which existing locomotor features are used.

The divergent effect of ACV on the probability of HL states inside and outside the odor-zone is consistent with our previous analysis (*Jung et al., 2015*) and shows that the effect of ACV can be described by the change in the probability of the fly occupying HL states. To assess whether there is a more fine-grained spatial structure to the effect of ACV on a fly's behavior, we divided the arena into a 60-by-60 grid and measured the ACV-induced changes in the probability of occupying a given HL state at each of the 3600 locations (*Figure 6*). The probability that a fly is in HL state 1 increases dramatically only at the edge of the odor-zone (*Figure 6A*), and not throughout the odor-zone where the odor concentration is uniform throughout, showing that the effect of odor on locomotion has a fine-grained spatial structure. Location-specific change in the probability of each HL state are shown in *Figure 6B*. The fine-grained modulation of locomotion is observed in other states as well - increases in state 2 are largest in an annular region just inside the odor-zone and increases in state 3 are largest at the very center of the arena. A similar specificity is observed in the increase in the probability of HL states outside the odor-zone. Increases in the occupancy of state 8 are uniform across the entire chamber outside the odor-zone; in contrast, the occupancy of states 9 and 10 increases in the region close to the odor-zone.

The structure that we observe represents the time-average over the entire duration of the odor period, and ignores the time evolution of the behavior (*Figure 6—figure supplement 1* and discussion). We were unable to explore the spatio-temporal evolution of behavior because of substantial fly-to-fly differences in locomotion and how it is modulated by odor.

## Both locomotion and an odor's effect on locomotion is fly dependent

Given that ACV affects the occupancy of HL states, it should be possible to do the reverse, that is decode the presence of ACV based on the distribution of HL states. Surprisingly, a variety of different decoding techniques failed to decode the presence of ACV based on the distribution of HL

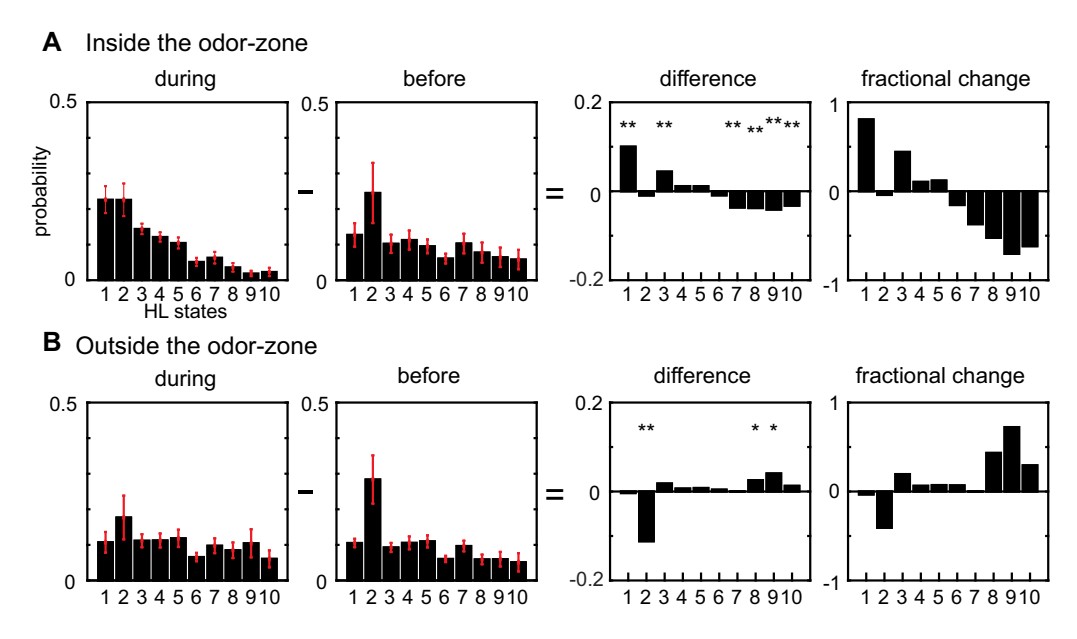

**Figure 5.** Odors modulate locomotion by altering the time spent in different HL states. (A) Odor-induced changes in the occupancy of HL states inside the odor-zone. Bar graphs showing the probability distribution of the 10 HL states during odor, before odor, the difference between the two and the fractional change in the distributions when the fly is inside. Red lines show ±bootstrapped 95% confidence intervals. Asterisks indicate significance at 0.05 with Bonferroni-Holm correction (**) and 0.01 (*) without correction based on bootstrapped hypothesis testing for equality of means. (B) Same as in A, but for outside the odor-zone.

DOI: https://doi.org/10.7554/eLife.41235.012

The following figure supplement is available for figure 5:

**Figure supplement 1.** For each HL state, the composition of LL states remain the same .

DOI: https://doi.org/10.7554/eLife.41235.013

states. One such method (*Figure 7—figure supplement 1*) in which we employed logistic regression to classify each one second of every fly's track into 'ACV present' or 'ACV absent' failed. Even more surprisingly, population decoding based on HL states did not perform any better than decoding based on the observables (*Figure 7—figure supplement 1*). One possibility that the logistic regression approach failed is because the average behavior represented in *Figure 5* does not accurately encapsulate individual fly behavior. Large fly-to-fly differences, where different individuals have fundamentally different basal locomotion or response to odor, might doom decoding methods aimed at discovering a single set of regressors that captures individual fly behavior.

Consistent with large fly-to-fly differences in behavior, we found that the distances between empirical flies are much larger than the distances between the synthetic flies (*Figure 7A*). Synthetic flies were generated as described in *Figure 7—figure supplement 7–2*. It is statistically impossible ($p < 10^{-131}$) that the observed Euclidean distance represents variations around the same average fly. The same conclusion applied to the fly's behavior in the other three conditions (before-inside, during-outside and during-inside: *Figure 7—figure supplement 3*).

Because the data in *Figure 7A* is inconsistent with individual flies being variations around a single average fly. We assessed whether the observed variability can be approximated based on a small number of discrete locomotor-types. X-means clustering (see Materials and methods section 6) showed that there are 4 clusters of flies based on their locomotion outside the odor-zone, before odor onset (*Figure 7B*). Although the identity of flies that cluster together changed, a similar number of clusters was found in each of the four conditions (Before odor/inside odor-zone, during odor/ inside odor-zone, before-outside and during-outside, *Figure 7* and *Figure 7—figure supplement 3*). Importantly, X-means clustering on a set of 34 randomly sampled points from a uniform distribution in the probability simplex space that the data reside in found no clusters. The Euclidean distances between synthetic flies drawn from the four different clusters were similar to the distances

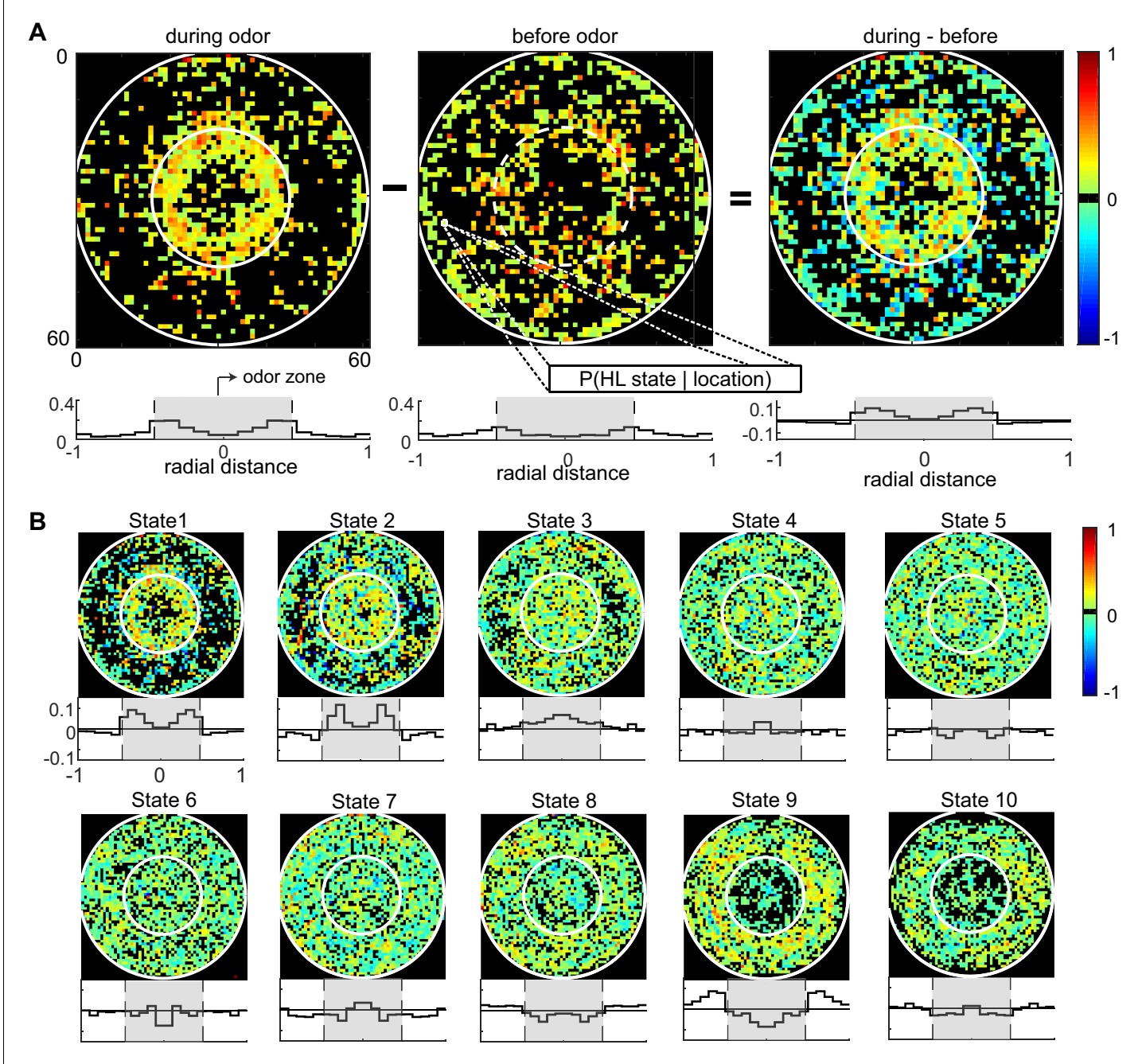

**Figure 6.** Fine spatial structure underlying odor-evoked changes in HL state occupancy. (A) The probability distribution of HL states before and during odor was calculated for each of the 60 × 60 bins. Taking HL state 1 as an example, each specific bin represents the probability of HL state 1 given the spatial location. White circles show the extent of the odor-zone and the arena. Bottom row: Probability of HL state 1 as a function of radial distance. The shaded region indicates the odor zone. (B) Change in HL state occupancy for each of the 10 HL states.

DOI: https://doi.org/10.7554/eLife.41235.014

The following figure supplement is available for figure 6:

**Figure supplement 1.** Temporal structure underlying HL state occupancy in response to odor.

DOI: https://doi.org/10.7554/eLife.41235.015

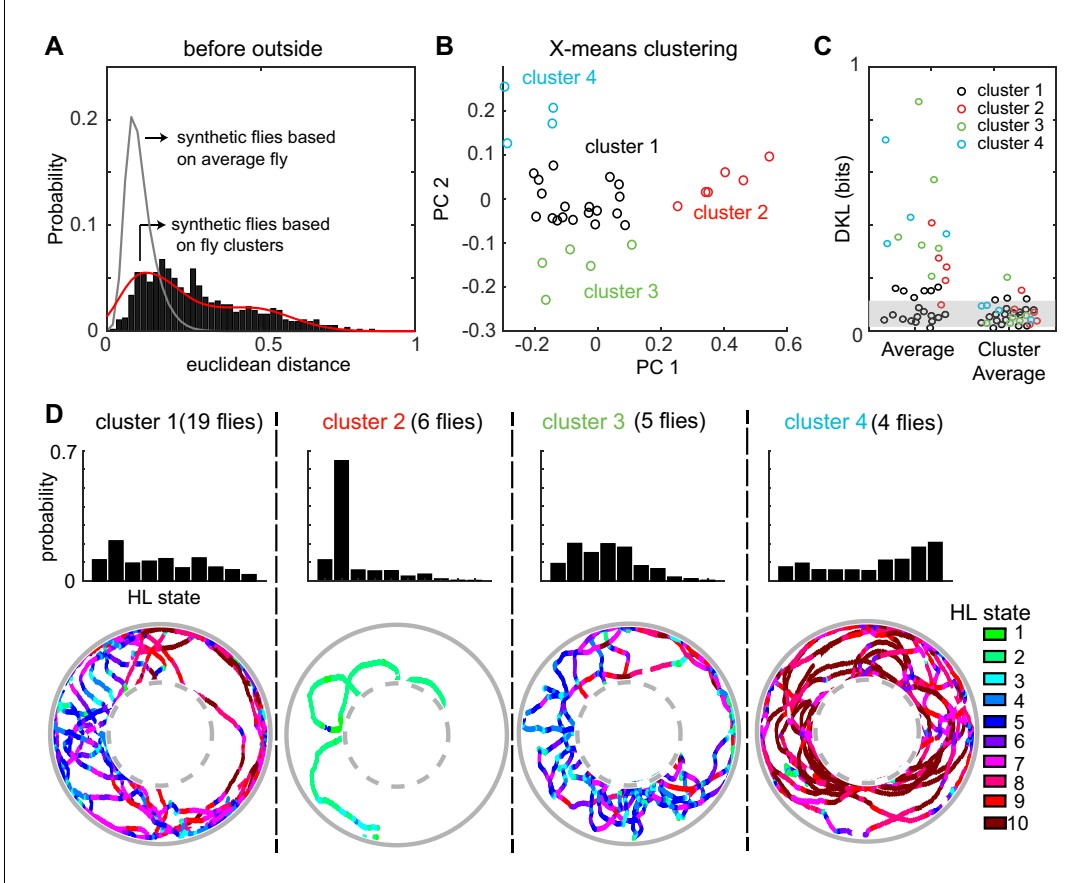

**Figure 7.** A fly's locomotion cannot be explained as variations around an average fly and is well-described on the basis of three to four locomotor types. (**A**) Histogram of distances between flies in the 10-dimensional space formed by the HL states. The distance between 100 iterations of 34 synthetic flies based on the average distribution of states is much smaller (gray line, Wilcoxon rank sum test, p < 10$^{-131}$). The distance between synthetic flies drawn from four clusters of flies (red line) has a distribution more similar to the empirical distribution. (**B**) X-means clustering (a variant of K-means) in the 10-dimensional HL state space. Only the first two PCs are shown. Each cluster is represented by a different color. (**C**) Left: KL divergence from the average fly to the individual fly (n = 34). Right: KL divergence from the average fly in each cluster to each individual fly in the corresponding cluster. Shaded area reflects the expected KL divergence (99% of KL divergences) from the HL state distribution of the average fly to the HL distribution of synthetic flies generated from this average (n = 3400). (**D**) Average HL state distribution for each cluster is shown in the top row. Bottom row shows an example fly for each cluster with HL states overlaid.

DOI: https://doi.org/10.7554/eLife.41235.016

The following figure supplements are available for figure 7:

**Figure supplement 1.** HL state distributions of the population are poor predictors for the presence of odor.
DOI: https://doi.org/10.7554/eLife.41235.017
**Figure supplement 2.** The process for generation of synthetic HL state sequences.
DOI: https://doi.org/10.7554/eLife.41235.018
**Figure supplement 3.** Flies can be clustered into three to four types based on their locomotion.
DOI: https://doi.org/10.7554/eLife.41235.019
**Figure supplement 4.** Cluster assignments are stable over time.
DOI: https://doi.org/10.7554/eLife.41235.020
**Figure supplement 5.** Locomotion before odor onset is only weakly predictive of locomotion during the presence of odor.
DOI: https://doi.org/10.7554/eLife.41235.021

between empirical flies (*Figure 7B* and *Figure 7—figure supplement 3*). Since X-means clustering tends to underestimate the actual number of clusters in data (*Pelleg and Moore, 2000*), the analyses in *Figure 7* and *Figure 7—figure supplement 3* suggest that there are at least three to four fly-types based on the frequency with which they use the HL states.

Consistent with the analysis with Euclidean distances above, the KL divergences (*Figure 7C*, left set of data points) show a large range indicating that some, but not all, flies are well-represented by the population average while others are not. Employing three to four fly-types defined as average distribution of their respective cluster decreased the information loss (*Figure 7C*, right).

How was the behavior of the flies in the four clusters different from each other? The average occupancy of the HL states for flies in each cluster and one example from each cluster is shown in *Figure 7D*. Cluster two was distinct from the others because flies move at markedly slower speed and spent >60% of their time in State 2 (*Figure 7D*) during which the fly was often stopped. Locomotion of flies in the largest cluster (cluster 1) was characterized by an alternation between medium- and slow-speed states. Flies in this cluster employed states 5 and 7 with a high frequency while making radially inward forays into the center of the arena. Similar behavior was observed in cluster 3, except that the flies employed the slower medium-speed states (states 4 and 5). Finally, the fourth cluster of flies demonstrated a different locomotor strategy. Flies in cluster four traversed the arena in concentric circles using the high-speed states (states 8–10). Thus, X-means performed on the HL state distributions appear to identify different locomotor strategies employed by the fly.

How similar is the locomotion of a fly at different times during a trial? To investigate this issue we first examined whether the mean behavior of flies from different clusters are different enough that they can be accurately clustered based on a small sample of the HL states (*Figure 7—figure supplement 4A*). We limited the analysis to before-outside and during-inside scenarios because the fly spent much of its time in these two scenarios. Only a one-second chunk of data is sufficient for better than chance clustering, and just 30 s of sampling is enough to accurately classify >85% of the flies into their respective clusters. We performed two analyses to test whether a fly's behavior is persistent: First, we divided the tracks into bins of different length, and asked whether cluster assignments based on small bin sizes is stable (*Figure 7—figure supplement 4B*). We found that state distribution within each bin was highly predictive of the cluster they belonged to. Second, we repeated the same analysis, but with bins of varying size starting from the first data point or ending at the last data point (*Figure 7—figure supplement 4C*). These analyses show that within the admittedly short timeframe of our experiments, the cluster assignments are stable.

Apart from the fly-to-fly variability, another reason why decoding based on the average fly fails is that the behavior of the fly before odor onset is only weakly predictive of its behavior during the presence of odor. Some flies exhibited similar behavior in the before odor/outside odor-zone, but were divided into separate clusters in the presence of odor because their locomotion differed (e.g. flies 11 and 33, and 17 and 27; *Figure 7—figure supplement 5A*, see the distributions of HL states). A similar trend is observed inside the odor-zone (*Figure 7—figure supplement 5B*). These examples imply that behavior before odor onset is unlikely to be strongly predictive of behavior during the presence of odor. This conclusion is supported by the weak correlation between Euclidean distances between pair of flies in the before and during periods (*Figure 7—figure supplement 5A* and *Figure 7—figure supplement 5B*).

## A small number of strategies can explain the variability in flies' response to odor

The analysis presented above suggests that individual differences explain why the logistic regression approach based on the average HL state distribution across flies failed to decode the presence of ACV from its absence based on the HL states usage by individual flies. If so, individualized logistic regression should be more successful. Logistic regression based on individual flies to decode the presence or absence of odor based on HL state occupancy during a 1s-interval was able to classify odor-no odor at better than chance level for every single fly (*Figure 8A*, see Materials and methods section seven and *Figure 8—figure supplement 1* for details). Moreover, as expected, logistic regression using the HL states performed significantly better than did the observables (*Figure 8B*) which indicate that HL states are more predictive of the presence of ACV than the observables.

The analysis in *Figure 8A* shows that the occupancy of HL states is predictive of the presence or absence of odor when the analysis is performed at the level of individual flies. Does this mean that each fly follows an individualistic strategy? To evaluate whether a fly's response to ACV cluster into a small number of response types, we once again started with X-means clustering based on the change in state occupancy before and during odor. X-means clustering found five clusters inside the odor-zone and four clusters outside the odor-zone (*Figure 8C*). Using these clusters as a starting

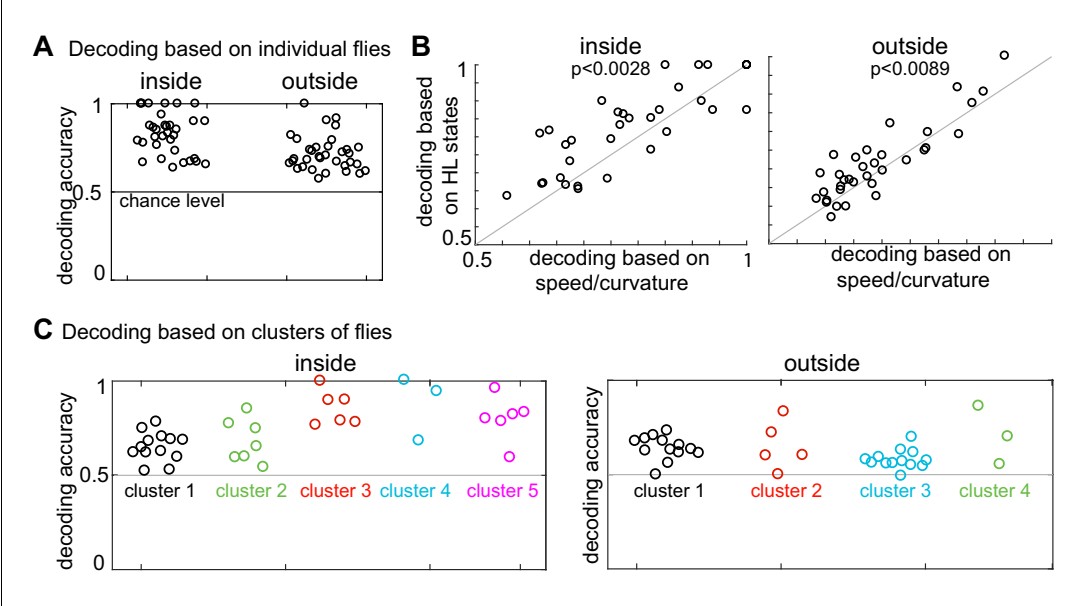

**Figure 8.** Presence of odor can be decoded based on both HL state distribution of individual flies and clusters of flies. (A) HL states are a good predictor of whether a fly is experiencing odors. Results of a logarithmic regression (logit) model fit to individual flies consistently shows better probability of correct predictions over chance (grey line). (B) HL states have higher predictive power than speed/curvature for individuals. Logit fits based on HL states were compared to that fit to the speed and curvature for inside (left) and outside (right) of the odor ring. A two-sided Wilcoxon signed rank test showed improved decoding using the HL states (p < 0.0028 for inside and p < 0.0089 for outside).(C) X-means clustering (a variant of K-means) based on the 10 dimensional space formed by the HL states show that there are five clusters of flies for difference in the HL state distributions inside and four clusters of flies for difference in the HL state distributions outside. Results of the logistic regression model fit to these clusters for the inside and outside cases are shown. Each cluster is represented by a different color.

DOI: https://doi.org/10.7554/eLife.41235.022

The following figure supplement is available for figure 8:

**Figure supplement 1.** Schematic showing how logistic regression for a single fly was performed.

DOI: https://doi.org/10.7554/eLife.41235.023

point, we could reconfigure the clusters such that the logistic regression on flies in each cluster performed at a better than chance level for each fly in the cluster (*Figure 8C*, see Materials and methods section seven for details), thus implying that a fly's response to odors can be approximated as a choice between few response-types.

Based on their behavior inside the odor-zone, the flies were divided into five clusters, four of these clusters have more than three flies (*Figure 9A*). Flies in cluster 5, just like the average fly (in *Figure 5*), slow down inside the odor-zone. Flies in cluster 3 also demonstrate a strategy similar to flies in cluster five except that ACV causes a large decrease in the time a fly spends in the medium-speed states rather than the high-speed states. In contrast to clusters 3 and 5 during which the fly slows down inside the odor-zone, flies in cluster two demonstrate a fundamentally different strategy in which there is a large decrease in state 2 in favor of states 1, 3 and 4. The flies in this cluster go from stop-start locomotion to locomotion in which they either meander at slow speeds or walk slowly with many sharp turns. Finally, for the flies in cluster 1, there is no dramatic change in state. These different strategies represent diametrically different effects of ACV on some HL states – the most striking example is the opposing effects of the odor on HL state 2 occupancies in different clusters – a large decrease in cluster 2, and an increase in clusters 3 and 5. These differences explain the odor-induced increase in usage of all the slow states in the average fly inside the odor-zone, except state 2 (*Figure 6A*).

Based on how odors modulate their behavior outside the odor-zone, there were four clusters. Behaviors that represent three of these clusters are shown in *Figure 9B*. Flies in cluster 1 decrease the time they spend in slow-states (state 1–2) and instead spend time in the fast states (states 8–10) resembling the behavior of the average fly. Cluster 2) shows a large decrease in HL state 2

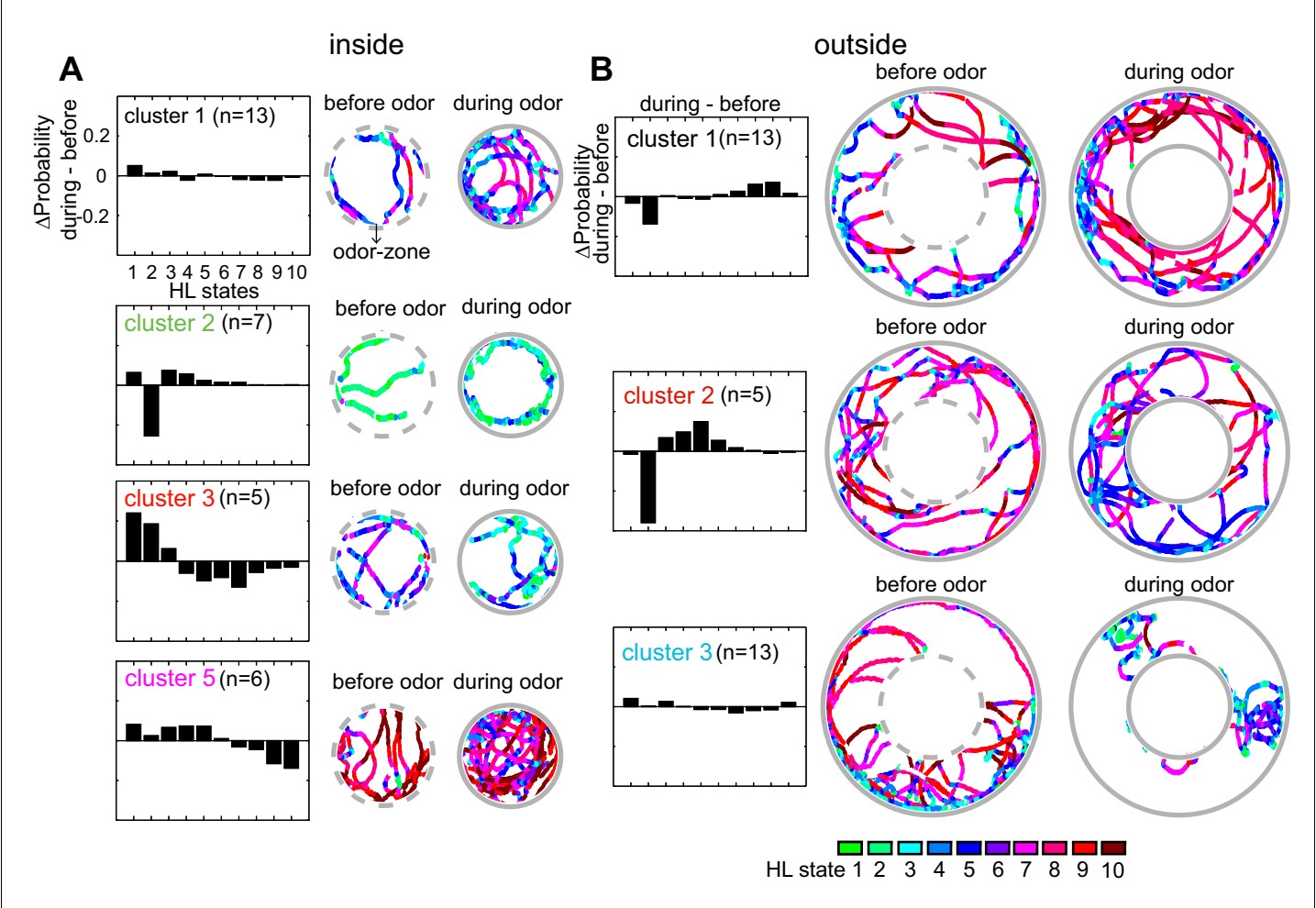

**Figure 9.** Flies' response to odors cluster into a small number of distinct, sometimes opposing response-types. (**A**) The difference between HL state distributions (during-before) and tracks from an exemplar fly from each cluster is shown. Dotted grey circle indicates the odor-zone. (**B**) Same as A, but for odor-evoked changes outside the odor-zone.

DOI: https://doi.org/10.7554/eLife.41235.024

occupancy similar to the behavior of flies in inside odor-zone cluster 2 while exhibiting a large increase in medium-speed states (state 4–6). Cluster 3 showed no dramatic change in state.

## Discussion

A cornerstone of neuroethology is that behavior unfolds in discrete packets, that is, behavior can be temporally segmented into natural units (*Barlow, 1977*; *Tinbergen, 1996*; *Baerends, 1976*). In some behaviors, these discrete packets are readily recognizable. But, in most daily behaviors, there is enough variability in these discrete packets to make the underlying natural units unrecognizable without the help of sophisticated analytical tools. Recently, there has been significant progress in discovering structure in sequences of posture in *C. elegans* (*Stephens et al., 2010*), *Drosophila* (*Berman et al., 2016*; *Vogelstein et al., 2014*) and mice (*Wiltschko et al., 2015*). Here we describe structure in a fly's locomotion in the velocity space.

Our salient results are:

- A fly moves at a relatively constant $\hat{v}_{\parallel}$ and $\hat{v}_{\perp}$ for an extended time that can last tens of steps. Therefore, a fly's locomotion can be decomposed into a few locomotor features – 10 features in the case of the model we present. The same 10 locomotor features could describe the

behavior of all flies in our dataset. The effect of odors on locomotion, and the differences in behavior across flies can be described in terms of these 10-locomotor features.

- Using this analytical framework, we show that odors affect locomotion by altering the time that a fly spends performing a given locomotor feature (instead of creating new features). The odor-induced change in locomotion has a fine spatial scale – the fly's response to odor changes as it moves from the border of the odor-zone to its center, and as it moves away from the odor border.
- The HHMM framework also allowed us to show that flies used the same 10 locomotor features, but in different proportions. The variation is so large that the fly's behavior cannot be understood as variations around the same average fly. Instead, the flies employ a minimum of at least 3–4 different strategies.

Below we discuss the limitation and implication of these findings.

## Model limitations

The model presented here is *a* model of locomotion and not *the* model of locomotion. The choice of observables and model strongly influences the features of the structure that is discovered. Our particular model reveals the structure of locomotion in the velocity space.

In choosing the observables, we employ a common method for describing locomotion, that is we treat the fly as a point object and measure the instantaneous change in the position of this point object; therefore, much of the insights from the model relate to how the fly changes its position in time. Apart from $\widehat{v}_{\parallel}$ and $\widehat{v}_{\perp}$, another similar and more commonly used representation of the change in fly's position: instantaneous speed and angular speed yielded similar locomotor features (data not shown). Ultimately, we used $\widehat{v}_{\parallel}$ and $\widehat{v}_{\perp}$ because this representation is more closely related to movement representation within the insect brain (*Green et al., 2017*; *Heinze, 2017*; *Turner-Evans and Jayaraman, 2016*), and because the measurement errors associated with angular speed are particularly large when flies moves slowly (*Gallagher et al., 2013*).

A fly's position can also be described using the actual position of the animal as observables rather than the change in the position, as employed in a recent study in rats (*Shan and Mason, 2017*) Using the instantaneous position as an observable would reveal different aspects of the structure underlying an animal's locomotion. Consider the trajectories of flies in Clusters 1 and 3 (*Figure 7*). They cross similar spatial positions, but are classified into different states because the flies travel at different speeds. In terms of the sequence of position in space, flies in both clusters have a similar behavior – they explore the outer arena border and make occasional radial forays inside the odor-zone. An analysis based on position would likely place these two clusters of flies together whereas our analysis, in the velocity space, placed them in different clusters.

Model architecture is also important. A hierarchical model performed better than a non-hierarchical model. The current model has state durations of <3 s. It is clear to human observers that there is structure in the data that is >3 s long. Flies sometime explore the outer border of the arena using characteristic paths that can last up to a minute. The short duration of states in our model cannot capture structure on these long-time scales. One possibility is choosing a deeper-layered architecture. Given the structured transitions between the HL states in our model, it is likely that if we used a deeper-layered architecture, we would likely uncover structure on a longer timescale.

## Locomotor features and implications for neural control of behavior

During both HL states 1 and 2, the fly's locomotion is quite slow, but in state 2 the fly stops and runs intermittently while in state 1, the fly is continuously in motion, albeit slowly. Similarly, in each of the HL states 4, 5 and 7, $\widehat{v}_{\parallel}$ lies within a narrow range, which is distinct for each of these three states, implying a tight control over forward speed. These locomotor characteristics can persist over 3 seconds (*Figure 2—figure supplement 2*) – a time period during which a fly takes 30 steps on average (given a step frequency of 10 Hz; *Mendes et al., 2013*). This tight control over $\widehat{v}_{\parallel}$ over many steps strongly suggests that locomotion unfolds in blocks. The HHMM presented here or the states revealed by it may not reflect the actual states employed by neurons in the brain. In fact, there is an ongoing debate whether behavior and its control is better represented as a continuum than by discrete states. The presence of long-lasting states that are employed repeatedly by all flies in our dataset implies that either locomotion does consists of transition between discrete states, or that these

states represent fixed points or peaks of a dynamical system around which the animal spends most of its time (*Berman, 2018*).

Another surprising result is that the same set of locomotor features describes the behavior of all the flies in the dataset. This result is particularly surprising given that our model explicitly allows each fly its own set of locomotor features. The fact that all flies can be reasonably modeled by the same model implies that within a given environment all flies construct their locomotion from the same building blocks, and differences in locomotion amongst flies or the effect of sensory stimulation can be quantified as changes in the frequency with which these building blocks are employed. An important avenue for future research is to assess whether these locomotor features are fixed or flexible.

## Odors affect behavior on a fine spatial scale

In nature, animals encounter odors in a cluttered and dynamic sensory environment (*Riffell et al., 2014*). Discriminating between odors, navigating towards the chosen odor source and pinpointing the odor source requires a flexible deployment of multiple different motor programs. It is difficult to replicate the complex natural environment in the laboratory. Therefore, laboratory studies are typically aimed at different subsets of the complex environment experienced by animals. In insects, much research has focused on an environment in which it experiences odors in a highly structured odor plume often within a high-contrast visual environment (*Budick and Dickinson, 2006*; *Kennedy, 1983*; *Vickers, 2000*; *van Breugel and Dickinson, 2014*; *Álvarez-Salvado et al., 2018*). Recently, similar experiments have been repeated for flies walking towards an odor source (*Bell and Wilson, 2016*). These experiments model an insect's behavior under one specific condition wherein the fly tries to locate an odor source at a distance using strong directional information from wind and vision. The experiments described here explore a fly's behavior in a small, dark circular arena. At most locations in the arena, the air speed was 0.07 m/s; the highest wind-speed was 0.11 m/s (*Jung et al., 2015*), a value lower than has been employed in most studies. Therefore, non-olfactory directional cues from vision or wind are minimized (*Jung et al., 2015*). Consistent with this idea, there was no change in the distribution of the flies when wind was completely eliminated.

We find that this behavior near the odor source can be described by changes in the HL states. The clearest evidence that changes in HL states are a good description of the fly's behavior is the analysis in which we measured the spatial distribution of odor-evoked changes in HL states (*Figure 6*), and observed a pattern that strongly resembles the odor-zone. This analysis shows that the HL state description is accurate enough to facilitate discovery of arena structure. However, because we averaged HL state distribution over the entire 3 min of odor exposure, the analysis misses some details. The flies first detect odor slightly outside the odor-zone as defined in this study (*Jung et al., 2015*), and their behavior during the first 10 s after odor encounter differs from their behavior during the rest of the odor period (*Figure 6—figure supplement 1A*). Moreover, at least some of the spatial structure results from the change in the radial density of the fly as a function of time (*Figure 6—figure supplement 1B*). A fly's interaction with odor is dynamic and that HL states are a good analytical framework to extract the spatiotemporal patterning of fly's behavior by odor. This spatiotemporal pattern likely differs among flies; a full description of this pattern requires a larger dataset and represents an important avenue for future research.

## Flies show considerable variability in locomotion despite employing the same locomotor features

Even single-cell organisms and animals with simple nervous systems display substantial individuality (*Jordan et al., 2013*; *Gallagher et al., 2013*). Animals with larger nervous systems are likely to display even greater individuality, in the case of adult flies this individuality was demonstrated in the context of locomotor handedness (*Buchanan et al., 2015*) in a choice assay. The nature and extent of individuality is harder to assess in more complex behaviors because of the difficulty in assessing individuality in a large behavioral space; differences in behavior can simply be different instantiation of the same behavior, or reflect fundamental differences in behavior. In this study, we find that different flies employ the same locomotor features but use them in vastly different proportions. The observed variability between the flies is inconsistent with a single type of locomotor behavior but can be approximated by invoking 3–4 clusters of flies. Because the clustering framework we

employed (X-means) underestimates the number of clusters, and because there were only 34 flies in our dataset, it remains to be seen whether there are a few locomotor-types or a whole continuum of locomotor-types. Another limitation of this study is that we have not yet ascertained whether a fly's behavior persists over a longer time frame. Despite these limitations, this study makes two important contribution to the study of individuality. First, we develop a statistical framework to study complex behaviors. This method can be extended to examine whether the differences we observe truely represent individuality. Second, in many behavioral studies, researchers focus on the effect of some stimuli on behavior and make conclusions based on the average fly. In this study, we provide a framework for testing whether the description based on an average fly is appropriate, and ways to proceed if such an approach is inadequate.

The diversity of odor responses observed here is consistent with work done on moths where (*Willis and Arbas, 1998*) but in sharp contrast to recent work on walking Drosophila (*Bell and Wilson, 2016*) in which the authors reported that attraction to odors results from a stereotypical motor pattern. The authors of that study claimed that the relatively simple response to odors in their study is likely a result of their simple behavioral arena. Another possibility is that in their study there is a strong, directional wind cue. In the presence of a steady wind cue in a narrow arena, it is likely that the flies' locomotor behavior is dominated by upwind walking and suppresses other elements of their behavior. It is well-established that a fly's response to odor is strongly influenced by context, as was demonstrated recently by comparing the response to odors in different visual and air flow conditions (*Saxena et al., 2018*).

When considered from the viewpoint of an individual animal, this variability is hard to understand: A hungry fly in search of food should respond with a singular, hardwired behavior which represents an optimal strategy for locating food. However, species evolve as large populations of individuals, and a successful species should be able to adapt to fluctuating environmental condition. Recent work has shown that - bet hedging - a process by which the same genotype shows considerable phenotypic variation is important for adaptation to fluctuating environments (*Kain et al., 2015*). Having a diversity of phenotypes ensures that some individuals would thrive in any condition and behavioral variability is a feature not a bug and its careful consideration is critical.

Studying individual behavioral responses is also important to understand the mechanism underlying both the control of locomotion and how odors control locomotion. Analyzing behavior at the population level can provide important insights into an animal's response but does not provide the resolution necessary for understanding the neural mechanism underlying the moment-by-moment control of behavior at the level of individual flies. In this context, it is instructive to take a closer look at the average response to odors in the light of clusters of response to odors. The average response (*Figure 5A*) was surprising: the occupancy of all the slow states except state 2 is increased. The lack of increase in occupancy of state 2 results from a cluster of flies in which the occupancy of state 2 is strongly decreased (*Figure 9A*). A similar effect is observed in the response of the average fly to the medium-speed states – states 4–7. The occupancy of these states decreases in some flies and increases in others. Thus, the average fly is an aggregate of these different clusters of flies, each of which has a distinct response to odor. Disaggregation is an essential first step to understanding neural control of behavior.

## Materials and methods

All the code required to fit the HHMM model, and perform all the analysis in this manuscript can be found on Github (*Tao et al., 2018*; copy archived at https://github.com/elifesciences-publications/HHMM). The dataset can be found on Dryad (DOI: 10.5061/dryad.m930f2m).

### Collection of behavioral data

The methods used to collect the behavioral data were reported in a previous study (*Jung et al., 2015*). Briefly, flies were raised in a sparse culture. Flies that were 3–5 days post-eclosion were starved for 14–18 hr. Locomotion of a single fly was recorded for a 3 min period before odor was introduced (before period) and a 3-min period during odor (during period), using a video camera at a rate of 30 frames per second. The coordinates of the fly were extracted using a custom MATLAB program (https://github.com/bhandawat/fly-walking-behavior/tree/master/

tracking; *Bhandawat, 2017*; copy archived at https://github.com/elifesciences-publications/fly-walk-ing-behavior).

## Extracting observables from the trajectory

The behavioral arena was normalized to a unit circle centered at the origin. The raw coordinates of the centroid of the fly were smoothed using wavelet denoising followed by a locally weighted (low-ess) filter. Speed and curvature were defined exactly as in the previous study.

To quantify the behavior of the 34 flies, we computed the speed of the fly along the original direction of movement ($\widehat{v}_{||}$) and the speed of the fly perpendicular to the original direction of movement ($\widehat{v}_{\perp}$). $\widehat{v}_{||}$ at time $t$ was defined as the component of the velocity at time $t$ in the direction of velocity of the fly at time $t-1$. $\widehat{v}_{\perp}$ was defined as the component of the velocity at time $t$ perpendicular to velocity of the fly at time $t-1$ (*Figure 1C*). These values were calculated as follows:

$$\widehat{v}_{\perp}(t) = \frac{dy_{t-1}dx_t - dy_t dx_{t-1}}{\sqrt{dx_{t-1}^2 + dy_{t-1}^2}}$$

$$\widehat{v}_{||}(t) = \frac{dx_t dx_{t-1} + dy_t dy_{t-1}}{\sqrt{dx_{t-1}^2 + dy_{t-1}^2}}$$

Values of $\widehat{v}_{||}$ and $\widehat{v}_{\perp}$ found to be further than 4 standard deviations away from the average were set to values drawn from a normally distributed distribution ($\sigma = 1$) centered at the 4 standard deviation mark. The resulting $\widehat{v}_{||}$ and $\widehat{v}_{\perp}$ were then set as the observables used in fitting a 2-level Hierarchical Hidden Markov model (HHMM).

We employed $\widehat{v}_{||}$ and $\widehat{v}_{\perp}$ instead of speed and curvature because curvature is very noisy at low speeds because the calculation of curvature requires division by the third-power of speed. $\widehat{v}_{||}$ and $\widehat{v}_{\perp}$ are directly related to speed and curvature as follows

$$Speed = \sqrt{\widehat{v_{\perp}^2} + \widehat{v_{||}^2}}$$

$$Curvature = \frac{\widehat{v}_{\perp}}{dx_{t-1}^2 + dy_{t-1}^2}$$

## HHMM modeling

HMMs are widely used in a variety of fields for modeling time series data. An HMM is a Markov model which assumes that a given sequence of observations may be explained by a set of states that are not observed (or hidden states), and the time independent probability of transitioning between these states. The model processes which produce the observations in an HMM are hidden to the researcher and thus, the goal of fitting an HMM is to uncover the highest likelihood probability model parameters that can generate the data. Baum and others developed the core theory of HMMs (*Baum and Petrie, 1966*). Since then there has been much exploration of model architecture, and fitting procedure.

HMMs have been shown to be effective in modeling behavior because instantaneous measures of an observable are variable; therefore, behavioral states inferred by the application of simple thresholding to instantaneous measures of the observables are likely to be erroneous. HMM remedies this problem by inferring states based not only on the value of the observable at the current time point but also on the previous and following time points, and allows a more accurate determination of state (this idea is well-explained in *Figure 2* in ref 17). Specifically, the assumption of Markov dynamics with a sparse prior on state transitions penalizes the consideration of unlikely state transitions based upon recent history (forward filtering) and future destinations (backward smoothing).

The HHMM is an extension to the HMM which applies hierarchical structure in the form that higher level state is itself an HMM composed of its lower level states (*Fine et al., 1998*). The approach we take in exploring and fitting the HHMM closely follows the approach developed by Matt Beal in which he applied variational algorithms to fit HHMM to a time series of observables (*Beal, 2003*).

This section is divided into three parts. First is the description of the model, second is the details of the process by which the model is fit, and third is the thought process behind our model selection.

## Model description

The model we describe here has 10 hidden states at the higher level (HL) and 5 hidden states at the lower level (LL) (Figure 2A). The empirical data to which the model is fit is a time series of the two observables - $\hat{v}_{\parallel}$ and $\hat{v}_{\perp}$. In building the model, the aim is to assign each instant in this time series to a LL and HL state. Each LL state is associated with a joint probability distribution on the observables. Each HL state is described by the transition probability matrix of its LL states. Therefore, while fitting a HHMM, we are determining three sets of quantities: First, the distribution of $\hat{v}_{\parallel}$ and $\hat{v}_{\perp}$ which describes each LL states (Figure 3-S1A/B). Second, the transition probabilities (TP) between the LL states which describe each HL state (Figure 2-S3B). Finally, the transition probabilities between the HL states (Figure 2-S2A). Based on the values of $\hat{v}_{\parallel}$ and $\hat{v}_{\perp}$, the model assigns a sequence of HL and LL states which best describes the data.

TP matrix for the HL states and the LL states associated with each HL states is shown in *Figure 2— figure supplement 3*. In each case, the TP matrix describes the probability ($P_{ij}$) that a fly in a given state detailed in the $j^{th}$ column will transition to a different high-level state detailed in the $i^{th}$ row. The high-level states are arranged in ascending order of mean speed/variance in curvature ratio. Because of this arrangement, the TP for the HL states appears well-structured. From any state, there is a strong tendency to transition to one of the neighboring state. This tendency implies that flies transition to states which have a similar speed/curvature ratio. LL states that belong to the same HL state often have more similar speed/curvature ratios. Thus, there is less of a tendency for LL states to transition to the LL state with the closest speed/curvature ratio. Nevertheless, the LL state transition probability matrices, too, are sparse; signifying a distinct pattern of transition between LL states (*Figure 2—figure supplement 3B*).

*Figure 3—figure supplement 1A and B* shows the joint distribution for $\hat{v}_{\parallel}$ - $\hat{v}_{\perp}$ for each state. Each row represents one HL state and its low-level children. In each panel, the solid line represents the model predictions and dots represent empirical values. Each LL state is modeled as a multi-variate normal distribution of observables. The solid line represents the bound within which the model predicts 85% of the data points corresponding to a given LL state lie. The empirical data points represent randomly selected subset of data corresponding to instants which the model assigns to a given LL state. The close agreement between the two for most LL states implies that the model is an excellent descriptor of the observables. The few exceptions where either the model distribution is too broad or where the model is not a good descriptor of the data reflect cases in which there are not too many data points in the concerned state (See for example, LL state 1 for HL state 6, *Figure 3—figure supplement 1A* ). Each HL state is a composite of the 5 LL state because for every time instance the fly is in a given HL state, the model assigns a LL state. Therefore, we can consider the probability density function of a HL state as the sum of the probability density functions of its LL states. From this, we can calculate the $85^{th}$ percentile contour for the corresponding probability density function for the HL states (*Figure 3A2* , *Figure 4*). Again there is a strong agreement between model and empirical data.

## Fitting process

We will first formally define a HMM as follows:

| Variable | Description |
|---|---|
| $s_t \in [1,2,\ldots N]$ | Indicates which of the N states is occupied at time $t \in [1,2,\ldots T]$ |
| $A = (a_{ij})$ | A NxN transition matrix where $a_{ij}$ represents the probability of transitioning from state i to state j. |
| $O = o_1, o_2, \ldots o_T$ | A sequence of observables composing of the two-dimensional data set: $\hat{v}_{\parallel}$ and $\hat{v}_{\perp}$ |
| $B_{ti} = P(o_t \vert s_t = i)$ | Emission probability describing the probability of an observation $o_t$ $t \in [1,2,\ldots T]$ being generated from a state $s_t = i$; Assumed to be Gaussian. |
| $\pi = \pi_1, \pi_2, \ldots \pi_N$ | The initial probability distribution of states. |

For our two layered HHMM, we chose a structure composing of 10 high-level states (HL state) on the top level with each HL state being associated with five low-level states (LL states) on the bottom level (*Figure 2*). Each LL state is modeled as a multivariate normal (MVN) distribution on the observables, and the HL states as being fully described by the mixture of LL state distributions associated with the HL state.

Now we will define our two layered HHMM as follows:

| Variable | Description |
|---|---|
| $s_t$ [1,2,...N$_{HL}$] | Indicates which high level state is occupied at time t ∈ [1,2,...T] |
| $u_t$ ∈ [1,2,...N$_{LL}$] | Indicates which low level state is occupied |
| $Q = [qkl]$ | High level state transition probability matrix, $q_{kl}$ represents the probability of transitioning from high level state k to high level state l. |
| $A^k = (a_{ij}^k)$ | The low level state transition probability matrix. Here $a_{ij}^k$ represents the probability of transitioning from low level state i to low level state j given that the high level state is k |
| O = o$_1$,o$_2$,...o$_T$ | A sequence of observables composing of the two-dimensional data set: $\widehat{v}_{\parallel}$ and $\widehat{v}_{\perp}$ |
| B$_{tki}$ = P(o$_t$\|s$_t$, u$_t$) | Emission probability describing the probability of an observation o$_t$ t∈[1,2,...T] being generated from a HLS $s_t$=k and associated low level state $u_t = i$. Assumed to be Gaussian. |
| $\pi^0$ <br> $\pi^k$ | The initial probability distribution of HLS. <br> The initial probability distribution of LLS for HLS $k$. |

We can henceforth refer to the set of model parameters as $\theta = [\pi, A, B]$ and the latent state variables as $Z = [s_1, u_1, s_2, u_2, s_3, u_3 \ldots s_T, u_T]$. For a given set of observations, the goal is to obtain a posterior probability distribution over the parameters and latent state variables. We approximate this posterior distribution as a factorized distribution over latent variables and parameters, that is $p(\theta, Z) \approx q(\theta)q(Z)$, and use the Variational Bayesian Expectation Maximization (VBEM) algorithm to find the best approximation. $q(Z)$ is obtained using the forward-backward algorithm which provides sufficient statistics needed to update the approximate posterior distributions over the parameters. In this setting, posterior probability distributions over rows of transition probability matrices are assumed to be Dirichlet, and the prior is chosen to favor self-transition parameters, $\alpha_{ii} > 1$, while discouraging the use of unneeded states that is, $\alpha_{ij} < 1$ for, $i \neq j$. Initial state distributions were also assumed to be Dirichlet with $\alpha_i < 1$.

The emissions probability distributions associated with each state were assumed to be Normal inverse Wishart with a prior favoring zero mean and unit variance. For each computational run, the initial parameters of these posterior distributions were randomized.

The priors used for fitting were such that within state transitions (*i*-th HL state to *i*-th HL state) was six times higher than transitions across HL states ((*i*-th HL state to *j*-th HL state). The prior for all *i*-to-*j* transitions were same. The Dirichlet prior over state transitions which we used was very weak. Specifically, in strength, the prior we used corresponds to the equivalent of two observations relative to the >10,000 observations that we used to fit the model for each fly.

## VBEM algorithm

The Variational Bayesian Expectation Maximization algorithm functions by iteratively updating the two components of our factorized approximation to the true posterior over parameters and latent variables and exploits conditional conjugacy to identify explicit update rules for all distributions over parameters. The goal of the VBEM algorithm is described alternatively as minimizing model error as given by the Kullback-Leibler divergence between the approximate and true posterior distributions; or maximizing a lower bound on the marginal probability of the data given the model. It is perhaps easiest to see how the algorithm works by considering the KL(q,p) instantiation. Under our factorized approximation, this objective function takes the form

$$KL = -\int q(\theta)q(Z)\log p(Z, \theta|O)d\theta dZ + \int q(\theta)q(Z)\log q(Z)q(\theta)d\theta dZ$$

The VBEM algorithm minimizes this objective function using coordinate ascent in the q(θ) q(z) function space. That is we obtain iterative update rules by simply taking the functional derivative if the KL with respect to q(Z) for fixed q(θ) and then solving for q(Z). This results in the so-called E step

where we update the posterior distribution over latent variables by averaging the true joint distribution of observations, parameters, and latent variables over our current estimate of the posterior on parameters:

$$\log q(Z) \sim E_{q(\theta)}[\log p(O, Z, \theta)]$$

In the so called M step, the roles are reversed:

$$\log q(\theta) \sim E_{q(Z)}[\log p(O, Z, \theta)]$$

Here, the tilde indicates equality up to an additive constant. This two-step procedure is repeated until convergence. The simplicity of these equations belies the complexity of the actual calculation of the posterior distribution over latent state variables. This is accomplished using the well-known forward-backward algorithm. The particular implementation of the forward-backward algorithm used in VB differs from the traditional EM implementation in that the parameters of the transition probability matrix are obtained by exponentiation of the geometric mean of the transition probability matrices:

$$A_{ij}^* = \exp\left(E_{q(\theta)}\left[\log A_{ij}\right]\right)$$

Based on the current posterior over parameters instead of simply using a maximum likelihood or MAP estimate. Because we assumed the rows of the transition probability matrix to be sparse, this further encourages the model to leave un-needed states unused.

Regardless, based on the assignments of observable tracks to HHMM models, we can iteratively update model parameters for each cluster to maximize the probability of observing the observables given the model parameters and the initial LL state probability density using the forward-backward algorithm. This operation can be thought of as the maximization step where we are calculating the best set of model parameters ($\theta$) that maximizes the Q function.

Although the EM algorithm is guaranteed to get a better fit on every iteration, it often does not converge to a global optimum of the likelihood function. As such, we implemented a system of cluster pruning, splitting, and reassignment to perturb the system in the case of reaching a local minimal fit. In the pruning step, unused clusters are removed until we are left with at most two unused clusters. In the splitting step, one cluster was selected based on a pseudorandom selection weighted by the size of the clusters (number of flies best fit to the cluster). The flies are then clustered into two clusters using k-means clustering based on the expected distribution of lower level state usage. Individual HHMMs were fit to each of these new clusters. In the cluster reassignment step, we filled each unused cluster(s) with the fly(s) that had the worst fit to the current cluster assignment. After each step, we conducted 10 iterations of the EM algorithm. The sequence of perturbations was conducted for 10 iterations before a final fit using the EM algorithm was conducted until convergence.

### Mixture Model

To account for differing search strategies that may be utilized by different flies we also fit a mixture of HHMMs to our multi-fly data set. The goal of this model is to identify a small set of different HHMMs that can describe our entire data set by clustering flies according to the similarity in their locomotion. In this context, two flies are said to behave similarly if the same HHMM provides a good description of their behavior. To model this scenario, we added another layer to our Bayesian model of the fly movement dataset. In this topmost layer, we instantiate a Dirichlet process which probabilistically assigns a label, $z_n$, to the $n^{th}$ fly. When $z_n = k$, it indicates that the $k^{th}$ HHMM under consideration governs the fly's movement. Therefore, in addition to identifying the posterior distribution over high and low-level states, we also infer a probability distribution over cluster assignments. These probability distributions are then used to determine how much we should weight each fly's movement data when updating the parameters of the different HHMM's.

### Model selection

In this study, we experimented with models with a varying number of HL states (6-16), and low-level states (4-6). One limitation of our fitting procedure is that it is not possible to compare models with a different number of states objectively, and thus the choice of model depends on the investigator. We chose a model with 10 HL states for two reasons: The most compelling reason is that different

model runs with 10 HL states produced results that were more similar to each other than did model runs with either lower or higher number of states. Another reason is that as the number of HL states in the model is increased, many HL states are sparsely used.

We think that the range 6–12 is a fairly narrow range. One important point is that within this narrow range of states, the state duration in all the models we tried were very similar.

One important limitation of our modeling approach (which we realized in hindsight) is that it is difficult to compare across models because everything changes slightly– LL states change, HL states change. We are currently revising the model to keep the LL states fixed so that the modeling approach would mostly focus on how the HL select the LL states. This process will make it much easier to compare across models.

## Comparison of HMM with HHMM

### Block clustering of HMM

For a given HMM fit, we may obtain the transition probability matrix (A). To obtain higher level structure from the states fit to a HMM, we may cluster the states based on their likelihood to transition to each other. One method of sorting and clustering the transition matrix into a block diagonal structure involves the use of information bottleneck formalism (*Berman et al., 2016*; *Tishby et al., 1999*). We used this method to search for a given K=10 clusters with an inverse temperature $\beta$ from 1 to 300 and time lag values of between 1 to 45 (33 ms to 1500 ms). We found that $\beta$ from 100-200 with time lags of 10-40 showed some of the same structure as the HHMM. We showed one solution using a time lag of 10 and a $\beta$ of 100 on an HMM fit of 50 states with 31 used states (*Figure 1—figure supplement 1* ).

### Bayesian model comparison

Given an HMM and an HHMM model with equal number of total states and observables O, we wish to calculate the probability of the HHMM model given the observations (P(HHMM|O)). Using Bayes theorem, we can calculate this with:

$$P(HMMM|O) = \frac{P(O|HHMM)P(HHMM)}{P(O)}$$

Utilizing a flat prior and Bayes factor:

$$K = \frac{P(O|HHMM)}{P(O|HMM)}$$

We now obtain:

$$P(HHMM|O) = \frac{1}{1 + \frac{1}{K}} = \frac{1}{1 + e^{-\log(K)}}$$

We utilized the evidence lower bound to approximate $P(O|HHMM)$ and $P(O|HMM)$. From this we can interpret the $P(HMMM|O)$ as 1 – p-values where $H_0$ = the HMM model is a better model fit. We obtain a $P(HMMM|O) > 0.9999$ indicating that we can reject the HMM model at p < 0.0001.

### Parameter comparison

To define an HMM, we need the transition probability($p(x_j \mid x_i)$ for and the description for the probability distributions defining each state. This means that there will be $K^2 + Kn_p$ parameters where $K$ designates the total number of states and $n_p$ designates the number of parameters defining the distributions. To define a HHMM, we need the transition probabilities at each level $\mathrm{p}(x_j|x_i)_l$ for $1 \leq i, j \leq K_l,\ 1 \leq l \leq L$ and the probability distributions defining each state. This means that in a model architecture in which each state at a level (*l*) is composed of equal numbers of states one level lower (*l* − 1), there will be $K_L^2 + K_L(K_{L-1})^2 + \ldots + K_2(K_1)^2 + n_p \prod_{l=1}^{L} K_l$ parameters. The distributions used in this report consisted of two-dimensional multivariate Gaussians, which are described by five parameters ($\mu_1, \mu_2, \sigma_1, \sigma_2, \rho$). The model that we employ in the manuscript has $10^2 + 10*5^2 + 50*5$ = 600 parameters.

## Post-hoc analyses based on the HHMM model

### Pre-processing of the model output

For each time point, the model assigns the probability that the fly is in each of the 10 HL states. We only included time instances where the model assigned a probability of >0.85 for one of the 10 states; this condition was satisfied for 81% of all data points (*Figure 2B*). Instances during which the flies transitioned from one HL state to another HL state and back to the same initial state in less than five frames (~170 ms) were reassigned to the initial HL state. Furthermore, instances during which the fly only spends one frame (~30 ms) in a certain HL state before changing states were removed. These corrections were done because such short transitions are likely to arise from noise in the observables rather than rapid transitions. Tracks of HL and LL states were extracted to compute the empirical distribution of the observables (gray dots in *Figures 3A* and *4* and *Figure 3—figure supplement 1A/B*).

Each HL state track was translated to begin at the origin by subtracting the position of the first point in the track. Then we rotated each track such that the fly is moving in the forward direction (positive y axis) at the start of the track. We considered the overall vector of the 10 frames (330 ms) before each track as the vector defining the fly's directional intent prior to starting a track. We defined the angle of rotation as the angle between this vector and the forward direction. These translated and rotated tracks allow us to visualize the distinct types of locomotion defined by the HL states (trajectories in *Figures 3C* and *4*).

### Sorting of the HL states

The model output numbers the HL state in a random order with respect to the underlying distribution of observables. To better understand the structure underlying transitions between HL states, we rearranged the states from low-speed-high-turn states to high-speed-low-turn-states by sorting the HL states of the model based on the ratio of their mean speed over the variance in curvature.

### Analysis of the effect of odors on the HL states

The nominal radius of the odor-zone defined by the radius of the odor tube was 1.25 cm (*Jung et al., 2015*). However, because there was some spread of the odor outside the odor-zone, the actual radius of the odor-zone was 1.5cm (*Jung et al., 2015*). A fly was considered to be inside the odor-zone when it was within the 1.5 cm radius. Although the odor was turned on 3 min following the start of the recording period, because the odor was only present inside the odor-zone, we considered the first time the fly entered the odor-zone after 3 min (first entry) as the start of the odor period. Based on a combination of the presence of odor and the location of the fly, we parsed the data into four categories: These were defined as inside odor-zone before first entry ($B_I$), inside odor-zone following first entry ($D_I$), outside odor-zone before first entry ($B_O$), and outside odor-zone following first entry ($D_O$). The standard deviation (SD) of the probability of HL states (P) in each scenario were calculated as follows:

$$SD_j = \sqrt{\frac{P_j(1-P_j)}{N-1}}$$

$$\forall j \in \{1, 2, 3, \dots 10\} \text{ HL state}$$

where N is the total number of data points in the scenario.

For each fly, we measured the probability distribution of HL states occupancy for each of our four scenarios. *Figure 5* shows the average probability distribution of these HL state occupancy with error bars denoting 95% confidence intervals calculated by bootstrap resampling using the BCa method (*Efron and Tibshirani, 1993*). We then conducted a bootstrap test for testing equal means in order to determine significance in the change in HL state distribution after odor onset for inside and outside the odor region separately (*Efron and Tibshirani, 1993*). Briefly, we first consider a given HL state probability of occurrence inside the odor zone for before (B) and after first entry (A). Our null hypothesis was that there was no change in the mean probability of observing the HL state. To test the null hypothesis, we first constructed populations F and G by translating populations A

and B respectively such that F and G share a common mean. We then drew bootstrap samples of flies F′ from population F and G′ from population G and calculate the test statistic as follows:

$$TS(F',G') = \frac{\overline{F'} - \overline{G'}}{\sqrt{\frac{\sigma^2_{F'}}{n} + \frac{\sigma^2_{G'}}{n}}}$$

$$\sigma^2_{F'} = \sum_{i=1}^{n} \frac{F'_i - \overline{F'}}{n-1}$$

$$\sigma^2_{G'} = \sum_{i=1}^{n} \frac{G'_i - \overline{G'}}{n-1}$$

We repeated this process 10,000 times to generate a distribution of the test statistic that we would expect from the null hypothesis. We then calculated the empirical statistic using the same formulation as the test statistic: TS(A,B). Using the distribution of test statistic and our empirical statistic, we conducted a two-tailed test with alpha = 0.05 and 0.01. As our data included multiple statistical tests (one for each HL state), we corrected for multiple comparisons by applying the Holm-Bonferroni procedure. This process was repeated for outside the odor zone.

## X-Means clustering

X-means clustering is an extension of K-means clustering. K-means clustering is an iterative algorithm that assigns data points to one of K groups based on the distance between points and the cluster centers; in most versions of K-means clustering, the number of clusters is specified by the user. X-means extends the K-means algorithm by computing the Bayesian information criterion (BIC) scores associated with a given K-means model fit, and, therefore allows a better assessment of the appropriate number of clusters in the data (*Pelleg and Moore, 2000*). Following Raftery et al. (*Raftery, 1995*), we computed the t-score based on the change in BIC given an increase in the number of clusters for N = 34 flies as follows:

$$\mathrm{BIC_K - BIC_{K+1}} \approx t^2 - \log(N)$$

In this case, the t statistic and the corresponding p values represent the likelihood that k-means with a given cluster size will do significantly better than one with smaller cluster size. We chose the maximum cluster K that fulfilled t > 3.86 (p < 0.05) and within 5% of the minimum BIC.

In the analyses in *Figure 7* and *Figure 7—figure supplement 3*, we clustered flies based on a 10-dimensional representation of the fractional time a fly spends in each of the 10 HL states. We also employ X-means clustering in *Figure 8* to cluster a fly's response to odors. In this analysis, the clustering was performed on odor-evoked change in HL state occupancy: For inside the odor-zone, $D_I$-$B_I$ was computed for each fly and was the input to the X-means. For outside the odor-zone, $D_o$-$B_o$ was computed for each fly and was the input to the X-means (*Figure 8*).

The clustering was performed on the 10-dimensional representation; but, for visual representation, we conducted PCA on the HL state distributions in each of the four scenarios to obtain a 2-dimensional representation of the clusters of flies (*Figure 7* and *Figure 7—figure supplement 3*). The first two principal components explain less than 90% of the variance.

The 10-dimensional HL state probability distributions used in *Figure 7* and *Figure 7—figure supplement 3* reside on a 9-dimensional probability simplex:

$$\{x \in \mathbb{R}^{k+1} : x_0 + \ldots + x_k = 1, x_i \geq 0, i = 0, \ldots, k\}$$

To assess whether the x-means clusters found in our data set are valid, we conducted x-means on a set of 34 points sampled randomly from the uniformly distributed 9-simplex space our data resides in. To sample from this simplex, we used a flat Dirichlet distribution marked by the following density function.

$$f(x_1,\ldots,x_{K-1};\alpha) = \frac{\Gamma(\alpha K)}{\Gamma(\alpha)^K}\prod_{i=1}^{K}x_i^{\alpha-1}; where\ \alpha = 1$$

where K = the number of dimensions. We found that x-means did not cluster these sampled data points into clusters.

## Logistic regression model

We employed logistic regression (logit) models (*Cox, 1958*; *Dobson and Barnett, 2008*) – a generalized linear model which can be used to describe the relationship between independent observations and a binary dependent variable. In this study, we analyzed whether the distribution of HL states in a one-second window is predictive of whether the fly is inside the odor-zone or outside it. Because of the difference in a fly's behavior inside and outside the odor zone, we separately performed this analysis inside and outside the odor-zone. We performed three different logistic regressions in this study: The analysis process is the same; the only difference is how flies were grouped. In the first analysis (*Figure 7—figure supplement 1*), a single regression was performed for all the flies. In the second analysis (*Figure 8A/B*, *Figure 8—figure supplement 1*), we calculated regressions individually for each fly so that there are 34 regressions –one for each fly. Finally, in *Figure 8C*, we calculated logistic regression individually for each cluster of flies – five clusters inside the odor-zone and four outside the odor-zone. The process for performing logistic regression is described below and illustrated in *Figure 7—figure supplements 1* and *Figure 8—figure supplement 1*.

First, we divide the data into 1 s time-bins. We also performed logistic regression on data subdivided into 0.33, 0.66, and 3 s bins (corresponding to 10, 20 and 90 data points) with varying amounts of time overlap and found no notable differences in model predictions. Because we want the chance prediction to be 50% in each analysis, bins were randomly removed from either the before or during case such that the total number of bins were the same for the before period and during period. Next, we performed principal component analysis (PCA) on the distribution to obtain a smaller number of uncorrelated variables. We considered the smallest number of principal components that cumulatively explained over 90% of the variance in our analysis.

The resulting principal components were used as predictors in fitting to a logarithmic regression model. We used the MATLAB built-in function 'glmfit.m' to implement fitting to a generalized linear model. For fitting to logit model, we used a binomial distribution (having experienced the odor or not) and the 'logit' criterion. The resulting logistic function based on the population data was used to predict if a fly was experiencing odor in any given 1 s bin.

To evaluate the predictive power of the raw observables (speed and curvature) on the behavior of the flies, the GLM was fit using the speed and curvature as predictors instead of the distributions of HL states. To compare the relative probability of correct predictions on for each fly between two different types of GLM fits ($M_1$, $M_2$), we considered the perpendicular distance ($\vec{D}j$) of these fits from the line of unity (indicating perfect correlation).

$$\vec{D}_j(M_1,M_2) = \frac{M_1(j) - M_2(j)}{\sqrt{2}}$$

$$\forall j \in [1,2,3,\ldots 34]$$

To determine whether HL states performed better than the observables, a Wilcoxon matched rank test was conducted on the 34 distances calculated for each of the model comparisons (*Figure 8B*, *Figure 7—figure supplement 1*).

As X-means was conducted on the average distribution of HL states for each fly in each scenario, the partitioning obtained will not necessarily reflect the optimal clustering of flies based on ability to distinguish if the fly has encountered odor or not given a smaller (1 s) time bin observation of HLS distribution. To better partition flies into clusters based on both the average difference across time and smaller snapshots of difference across the 1 s time frame, we took X-means as the first partitioning of flies. Then we took a total of 6 flies from the largest clusters (n > 10) that had the worst predictive power given GLM fits based on clusters. We then redistributed these flies into clusters in order to maximize the sum of the predictive power of individual flies across all clusters.

## Generation of synthetic flies

HHMM synthetic flies were generated based on the transition probabilities for each of the four scenarios ($B_I$, $B_O$, $D_I$ and $D_O$) separately. To generate a synthetic track, an HL state was first chosen based on the occurrence probability distribution of HL states. At this point, a new HL state was assigned for the next time instance ($x_i$) based on the HL transition probability matrix. Since the empirical flies spent variable time in the four scenarios, in the creation of the tracks we chose the median time spent. Therefore, each synthetic fly lasted until the total duration was reached for each scenario (*Figure 7—figure supplement 2*). 100 sets of 34 synthetic flies were generated for each of the four scenarios (*Figure 7A*, *Figure 7—figure supplement 3*). The resulting synthetic average distribution of HL states for each of the four scenarios was compared across the 100 iterations and showed high consistency between iterations.

## Prediction of clusters-based subsampling

Time points were sampled using four methods based on bin duration for each fly for the two scenarios with the most data ($B_o$ and $D_i$) separately (*Figure 7—figure supplement 4*). In method one, for a given scenario, segments of continuous repeated HL states were randomly sampled and stitched together until the duration of the time bin was fulfilled. In the second method, a window lasting the time bin was sampled from the HL states for the scenario. In method three, the segments were sampled starting with the first time point of the scenarios. In method four, the segments were sampled starting with the last time point of the scenarios and moving backwards in time. After sampling, the average HL state distributions were calculated for each time bin and the Euclidean distance from the distribution to the centroid of each X-means cluster for the given scenario were calculated. The closest cluster was compared to the X-means cluster assignment based on all time points. This process was repeated 100 times to generate a mean percentage of correctly labeled flies based on the subsampling duration. Chance was calculated as the probability of choosing a fly for a given cluster and being in the Voronoi cell of the cluster. This translates to:

$$E(X) = \sum_{i=1}^{K} x_i p_i$$

where $x_i$ is the probability of observing cluster i based on the number of flies in each cluster, $p_i$ is a weighting based on the size of Voronoi cell in the simplex space, and K is the total number of clusters.

# Acknowledgements

We acknowledge the members of Bhandawat lab and Gaby Maimon for critical comments on earlier versions of the manuscript. This research was supported by the National Institute of Deafness and Other Communication Disorders, and the National Institute of Neurological Disorders and Stroke of the National Institutes of Health (under grant numbers RO1DC015827 and RO1NS097881, respectively) (VB) and an NSF CAREER award (under grant number IOS-1652647) (VB).

# Additional information

## Funding

| Funder | Grant reference number | Author |
|---|---|---|
| National Institute of Neurological Disorders and Stroke | RO1NS097881 | Vikas Bhandawat |
| National Institute on Deafness and Other Communication Disorders | RO1DC015827 | Vikas Bhandawat |
| National Science Foundation | IOS-1652647 | Vikas Bhandawat |

The funders had no role in study design, data collection and interpretation, or the decision to submit the work for publication.

## Author contributions
Liangyu Tao, Conceptualization, Data curation, Software, Formal analysis, Validation, Investigation, Visualization, Methodology, Writing—original draft, Writing—review and editing; Siddhi Ozarkar, Conceptualization, Data curation, Formal analysis, Validation, Investigation, Visualization, Methodology, Writing—original draft, Writing—review and editing; Jeffrey M Beck, Conceptualization, Software, Formal analysis; Vikas Bhandawat, Conceptualization, Resources, Data curation, Formal analysis, Supervision, Funding acquisition, Validation, Visualization, Writing—original draft, Project administration, Writing—review and editing

## Author ORCIDs
Vikas Bhandawat (iD) http://orcid.org/0000-0002-2608-0403

## Decision letter and Author response
Decision letter https://doi.org/10.7554/eLife.41235.030
Author response https://doi.org/10.7554/eLife.41235.031

## Additional files

### Supplementary files
• Transparent reporting form
DOI: https://doi.org/10.7554/eLife.41235.027

### Data availability
Data has been deposited in Dryad Data Repository and is available at doi:10.5061/dryad.m930f2m. All the code required to fit the HHMM model, and perform all the analysis in this manuscript can be found on Github (https://github.com/bhandawat/HHMM; copy archived at https://github.com/elifesciences-publications/HHMM).

The following dataset was generated:

| Author(s) | Year | Dataset title | Dataset URL | Database and Identifier |
| --- | --- | --- | --- | --- |
| Tao L, Ozarkar S, Beck J, Bhandawat V | 2018 | Data from: Statistical structure of locomotion and its modulation by odors | https://dx.doi.org/10.5061/dryad.m930f2m | Dryad Digital Repository, 10.5061/dryad.m930f2m |

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
