## [Decision Letter]

Thank you for submitting your article "Statistical structure of locomotion and its modulation by odors" for consideration by *eLife*. Your article has been reviewed by three peer reviewers, including Ronald L Calabrese as the Reviewing Editor and Reviewer #1, and the evaluation has been overseen by K VijayRaghavan as the Senior Editor.

The reviewers have discussed the reviews with one another and the Reviewing Editor has drafted this decision to help you prepare a revised submission.

Summary:

In this manuscript, the authors present a creative analysis of fly walking in response to an attractive odor (apple cider vinegar, ACV) in circular arena in the dark and where odor can be strictly limited to a central zone. They monitor fly movement as velocity (parallel and perpendicular to prior movement) prior to and after odor is introduced to the arena center. They then analyze the data with a Hidden Markov Model (HMM) and a Hidden Hierarchal Markov Model (HHMM) and show decisively that the HHMM fits the data better and that the high level sates (HL) describe stereotypical locomotor components and that transitions between them vary in probability such that states with similar velocities have higher transition probability. They then show that introduction of odor affects the probability that a given HL state is occupied and varies inside and outside the odor zone with final spatial structure. A surprising finding is that the average probability distributions of the HL states cannot be used to determine the presence of the odor. Subdividing the flies according to their individual pre and during odor probability distributions and clustering reveal 3-4 categories of responses and there are different cluster of the 34 flies tested for each of the four conditions (pre inside odor zone, pre outside odor zone, during inside odor zone, during outside odor zone). The most interesting conclusions of the paper are: 1) the observation that the HHMM model fits better than the HMM model, implies that there is structure at multiple time scales in the data 2) under the specific conditions imposed that although all flies in dataset use the same set of locomotor features, individual flies vary considerably in how often they employ a given locomotor feature (HL state), and how this usage is modulated by odor. The paper should arouse general interest in the behavioral neuroscience community.

Essential revisions:

1) There are concerns about how well the data constrains particular HHMM put forward that can be allayed by putting confidence intervals on the probability distributions of Figure 5 and Figure 6.

2) Several of the reviewer concerns focus on dwell times in each HL state. These dwell time should be analyzed and distributions presented and interpreted.

3) The observation that the HHMM model fits better than the HMM model, implies that there is structure at multiple time scales in the data. For example, HL states may be prolonged because the systems transitions among nearby LL states within the HL state; is this supported by the dwell time of LL states within a prolonged HL state? The authors also note in the Discussion that there is structure on longer time scales. Can they think of some analysis that would pull this out?

4) The findings about individuality and the fact that the presence of odor can be predicted from a model that takes individuality into account but not from a model that does not are interesting. Here however, some additional analyses or data would help support the claim. For example, can the authors compare the distribution of states early versus late in the trial and show that each individual still occupies a characteristic set of states? How do HL state dwell times vary among individual or early vs. late in a trial? Would more flies analyzed and/or for longer period of time help resolve the individuality issues?

5) The detailed reviewer comments are appended and that should be addressed in light of the consensus concerns above.

*Reviewer #1:*

Concerns

1) Subsection “A small number of strategies can explain the variability in flies’ response to odor” third paragraph: be explicitly clear as to how many clusters were found for each case illustrated in Figure 9. In Figure 9A, I assume there were at least 5 clusters but that cluster 4 had less than 4 flies; were there other clusters? Be very clear as to the number of clusters for Figure 9A and 9B.

2) Results section: you state in results that flies "…spend 60% of time performing a locomotor feature for >300 milliseconds, and >10% of their time performing a single locomotor feature for >3s (Figure 2—figure supplement 1)”. The first statement appears compatible with the cumulative distribution in the figure, but I don't see the second at all. Am I missing something or is the maximum duration illustrated 1.5 s and less that 5% of states are of this duration or longer? Please plot in a supplemental figure the real distribution of time duration of states in the data. These data are essential if you are to make claims like "These locomotor characteristics can persist over 3 seconds – a time period during which a fly takes 30 steps on average (given a step frequency of 10 Hz). This tight control over 𝑣̂|| over tens of steps strongly suggests that locomotion unfolds, not on a step-by-step basis, but in blocks of tens of steps." Or like "A fly moves at a relatively constant 𝑣̂|| and 𝑣̂⫠ for tens of step." It seems for example, that for states that last 300 ms that only 3 steps are possible and there are many states that last shorter periods. Are moving states distributed in duration longer/shorter that stopping states?

3) Subsection “Odors affect behavior on a fine spatial scale.”, paragraph two: Here you are creating a sequence of behavior based on averages and YET you claim that average data does not describe a fly's behavior but there are distinct strategies. Please clarify.

*Reviewer #2:*

1) The model assumes that behavior is best modeled as set of discrete states (with the implication, I think, that this is how they are controlled neurally). The alternative possibility, alluded to briefly in subsection “Flies show considerable variability in locomotion despite employing the same locomotor features.”, is that some parts of the behavior could be better represented (or controlled) as continua. Since velocity is a continuous 1D variable, and since the fly must pass through intermediate velocities to transition from low to high velocity and vice-versa, I think this alternative should be considered. I am not suggesting the authors entirely revamp their model but I think this point could be discussed or considered a bit more prominently.

2) In the analysis of spatial structure (Figure 6), the role of time history should be considered/discussed. For example, behavior near the odor border might be different because the fly is more likely to have experienced no odor shortly before odor (or vice versa) than in the center of the arena. The responses of olfactory neurons are well known to show responses that depend on history over multiple time scales, so this point should be considered/discussed. Whether behavior could also be influenced by airflow at different parts of the chamber should be noted (without assuming the reader knows the details of the earlier paper). For example, could the airflow from the vacuum be causing the flies to slow down inside the odorized area, as in Yorozu et al., 2009?

3) The findings about individuality are quite interesting. As mentioned above, the analysis in which the presence of odor can be predicted based on individual flies or clusters of flies, but not the whole dataset is quite interesting. However, as the authors note in the response to Reviewer 3, individuality in responses has also been investigated elsewhere. A typical analysis in these studies (de Bivort, Branson) is to compare behavior of the same individual at different time points and to show that they are more similar to themselves than to each other. The authors might consider splitting their data in time and repeating the analysis, although the time interval of the data here is relatively short (6 min). In addition, it would be nice to know if they could correlate their clusters with anything else about the flies. In the response to reviewers the authors allude to different behaviors produced by different genotypes. I think it would add to the interest of the study if these data were included.

4) The comparison of HMM and HHMM models is nicely rigorous but seems highly technical for this journal. The authors should consider focusing the text on the conceptual conclusions that can be drawn from this comparison.

*Reviewer #3:*

Before publication some work is needed to address questions about: 1) how to interpret HL behaviors (survival time statistics of HL states and relationship to the structure of a 10HL-5LL model); 2) error bars.

Main comments:

Error bars.

I could not get a clear sense of how well the data is constraining the HHMM model. For example in Figure 5, what is the uncertainty of these distributions? Can the authors bootstrap the data? Likewise in Figure 6 how many data points go in each colored dot? What is the uncertainty of the probability plots as a function of radial distance? There are no error bars on these plots.

Time spent in HLs.

One of the reasons HHMM seems to work better than HMM is that behaviors have extended durations and since the trajectory can jump around the LL states within a single HL state, that HL state can last long. Even though this is a key aspect of the model, there is no analysis or plot of the waiting time distribution in each HL state that would give an idea of the "duration" of such states during behavior and how that duration depends on the number of HL and LL states used in the HHMM model. The only thing I could find was Figure 2—figure supplement 1 (but this includes all states) and the statement made: "A fly moves at a relatively constant 𝑣̂|| and 𝑣̂⫠ for tens of step. This tendency means that a fly's locomotion can be decomposed into a small number of locomotor features – 10 features in the case of the model we present.". However, this main conclusion is provided without plots about it. Supporting this point with quantitative data analysis is important because the authors use it to interpret various aspect of the results. It is also important to know if some HLs state last much longer than other in order to interpret the data. This may also clarify why 10HLs each with 5LLs are used in the HHMM. See below.

Subsection " HHMM reveals that a fly’s locomotion is surprisingly structured in the velocity space…".

An interesting outcome underlined by the authors (last paragraph) is that the resulting HHMM accounts for 80% of all the data (within 85% confidence interval) using 10 locomotor features (10 HLs). Some of the HL states correspond to clearly distinct behavior, such as 1=meander, 2=stop-to-walk or 9=fast right turn, 10=fast left turn. However, other HLs seem to be part of a continuum, such as 4,5,7= medium speed walking, and it is not entirely clear why 3 distinct HLs are needed to describe medium speed walking. Some HLs seem to fit less well the data, e.g. HL 6. I could not find a clear explanation of why 10HLs each with 5LLs were chosen and whether fewer HLs would have worked as well. What would happen if the HHMM had less HL nodes but more LL nodes in each HL? A discussion of survival times in each HLs might help sort this out and provide a way to interpret states 4,5,7. Are these different behaviors or one behavior distributed over 3 HL?

Subsection “Locomotor features and implications for neural control of behavior”.

In the discussion the authors say that v_parallel lies within a narrow range that is distinct for each three state and that 3 states reflect a tight control on locomotion by the brain. They mention these locomotor state can persist up to 3 second/30 steps and suggest that this indicates that the brain controls locomotion in blocks of 10 steps. I find this conclusion drawn from the finding of these 3 clusters too speculative given the data: In Figure 3—figure supplement 1 and Figure 4 the distributions of v_par overlap significantly between HL 4,5 and 7. This is again related to the two points above. Finally, if HL 4 5 7 are really distinct behaviors then another possibility could be that they are needed to account for fly-to-fly variability in the data, and that for a single fly only one state is sufficient to describe medium speed walk.

[Editors' note: further revisions were requested prior to acceptance, as described below.]

Thank you for resubmitting your work entitled "Statistical structure of locomotion and its modulation by odors" for further consideration at *eLife*. Your revised article has been favorably evaluated by K VijayRaghavan (Senior Editor), a Reviewing Editor, and two reviewers.

The manuscript has been improved but there are some remaining issues that need to be addressed before acceptance, as outlined below:

Significant additional revisions are still required. The comments of the expert reviewers, below, are detailed and require full responses.

*Reviewer #2:*

This manuscript compares HMM and HHMM approaches to clustering behavioral data from flies in an olfactory paradigm. Based on a rigorous comparison of models, the authors conclude that the HHMM model performs better than the HMM model, implying the presence of hierarchical long-time scale structure in the data. They further analyze the behavior of individual flies and show that responses to odor cannot be understood in terms of variation around an "average fly" but rather are better understood as belonging to 3-4 response types. Response types of individuals are stable over the duration of the experiment. Overall the methods introduced for analyzing complex behavioral data appear to be sound and are likely to be of broad interest to scientists studying natural behavior and its neural correlates.

Two points need to be addressed before acceptance:

1) How do the priors used in fitting constrain the structure of the transition matrices found for the models? In the Results section, the authors state that "the assumption of Markov dynamics with a sparse prior on state transitions penalizes the consideration of unlikely state transitions base upon recent history and future destinations." This suggests that a sparse prior was applied in fitting. However, later in the paper the authors state: "The transition probability matrix for the HMM was sparse, suggesting that from each state there are transitions to only a handful of other states." If a sparse prior was applied during fitting than the fact that the resulting transition matrix is sparse is not really a finding about the data.

Similarly, the authors need to clarify what fitting priors or constraints were placed in the HHMM model. The Results state "This is because any HHMM- which puts very specific constraints on the transitions probability matrix- can be represented by an HMM but not vice-versa." What are the constraints on the transition probability matrix for the HHMM? The text says "The transition probability matrix is sparse- a vast majority of transitions from each HL state were to 2-3 other HL states." Was this imposed by fitting priors?

2) Parts of the manuscript are somewhat long and repetitive. I think the manuscript could be productively shortened to have greater impact on its readers.

For example, in the section titled "HHMM reveals that a fly's locomotion is surprisingly structured in the velocity space" the last paragraph is devoted to restating the major conclusion of the first section: HHMM performs better than HMM. I think this could be reduced or folded into section one, which focusses on this comparison, and the section focused more exclusively on describing the HL states uncovered by the model.

In the Discussion subsection “Flies show considerable variability in locomotion despite employing the same locomotor features” paragraphs two and three are highly descriptive of the individual fly data and somewhat repetitive with parts of the Results subsection “Both locomotion and an odor’s effect on locomotion is fly dependent” (already rather long). I think these sections could be condensed to make the major points about individual variability in the Results, and use the Discussion mostly to compare these findings with the results of other studies.

*Reviewer #3:*

The authors have addressed most of my concerns expect the important one regarding the confidence interval on the distributions in Figure 5. What I am asking is for the authors to use bootstrapping to extract many sample distributions from subsets of the data in order to get a confidence interval on how these bins are populated by the data and how significant the changes are between before and during odor exposure.

---

## [Author Response]

Essential revisions:1) There are concerns about how well the data constrains particular HHMM put forward that can be allayed by putting confidence intervals on the probability distributions of Figure 5 and Figure 6.

There are two sources of variability. The first corresponds to how well the data constrains the model. This source of variability can be quantified as follows: The standard deviation (SD) of the probability of HL states (P) in each scenario can be calculated as follows:

SDj=Pj1-PjN-1

∀j∈1,2,3,…10HLstate

Where N is the total number of data points in each scenario. This standard deviation is extremely small and is shown in Figure 5.

The second source of variability is how different is the behavior of different flies in a given scenario. Getting to the bottom of this variability is the main question that drives the manuscript from Figure 7-9.

2) Several of the reviewer concerns focus on dwell times in each HL state. These dwell time should be analyzed and distributions presented and interpreted.

We have analyzed the dwell times of each HL state (Figure 2—figure supplement 2). Specifically, we plot the cumulative distributions of the HL states. We also plot the percentage of time a fly spends in a state of a given duration. These plots show the differences in dwell time in different HL state.

3) The observation that the HHMM model fits better than the HMM model, implies that there is structure at multiple time scales in the data. For example, HL states may be prolonged because the systems transitions among nearby LL states within the HL state; is this supported by the dwell time of LL states within a prolonged HL state? The authors also note in the Discussion that there is structure on longer time scales. Can they think of some analysis that would pull this out?

The plot on dwell time – Figure 2—figure supplement 2 – also plots the average dwell time of the low-level states and the number of LL state transitions/transition into HL state as a function of the duration of the HL state. These data support the idea stated above: The number of transitions increase when the duration of HL states become longer. Thus, this analysis works exactly as the reviewers envision above, and as one would expect if the data has structure on multiple time scales.

With regards to pulling out structure on an even longer time scale: we don’t think that there is any simple analysis that would work. We think that pulling out longer timescale likely requires a different model with more hierarchical layers, and likely a different set of observables.

4) The findings about individuality and the fact that the presence of odor can be predicted from a model that takes individuality into account but not from a model that does not are interesting. Here however, some additional analyses or data would help support the claim. For example, can the authors compare the distribution of states early versus late in the trial and show that each individual still occupies a characteristic set of states? How do HL state dwell times vary among individual or early vs. late in a trial? Would more flies analyzed and/or for longer period of time help resolve the individuality issues?

We performed the analysis suggested here (Figure 7—figure supplement 4). We find that even if you take a 1-second chunk of data, it assigns flies to the correct cluster at a level greater than chance. A 30-second chunk of data classifies >80% of the flies accurately. The correctly assigned fraction is about the same whether we use the first 30 seconds of data or the last 30 seconds of data.

It is important to note that what we are claiming is that these 34 six-minute trajectories do not belong to the same cluster. To establish individuality, one needs trajectories from the same fly over multiple days. It is also important to control the environmental conditions within a very tight bound. We were anal about the environment conditions. But, because the experiments were not specifically designed to get at the question of individuality, therefore the bounds on many conditions – age of the fly (3-5 days old), culture conditions (50-200 eggs) should be controlled even more tightly. Moreover, we did not control humidity at the time of the experiment. In essence, collecting longitudinal data under even more controlled conditions from a large population of flies is necessary to settle questions regarding individuality, and is well beyond the scope of this study.

5) The detailed reviewer comments are appended and that should be addressed in light of the consensus concerns above.

We agree with most of the reviewer critiques and have addressed them in the manuscript.

Reviewer #1:

Concerns1) Subsection “A small number of strategies can explain the variability in flies’ response to odor” third paragraph: be explicitly clear as to how many clusters were found for each case illustrated in Figure 9. In Figure 9A, I assume there were at least 5 clusters but that cluster 4 had less than 4 flies; were there other clusters? Be very clear as to the number of clusters for Figure 9A and 9B.

Thank you. We have specified the number of clusters in the figure in subsection “A small number of strategies can explain the variability in flies’ response to odor”.

2) Results section: you state in results that flies "…spend 60% of time performing a locomotor feature for >300 milliseconds, and >10% of their time performing a single locomotor feature for >3s (Figure 2—figure supplement 1)”. The first statement appears compatible with the cumulative distribution in the figure, but I don't see the second at all. Am I missing something or is the maximum duration illustrated 1.5 s and less that 5% of states are of this duration or longer? Please plot in a supplemental figure the real distribution of time duration of states in the data. These data are essential if you are to make claims like "These locomotor characteristics can persist over 3 seconds – a time period during which a fly takes 30 steps on average (given a step frequency of 10 Hz). This tight control over 𝑣̂|| over tens of steps strongly suggests that locomotion unfolds, not on a step-by-step basis, but in blocks of tens of steps." Or like "A fly moves at a relatively constant 𝑣̂|| and 𝑣̂⫠ for tens of step." It seems for example, that for states that last 300 ms that only 3 steps are possible and there are many states that last shorter periods. Are moving states distributed in duration longer/shorter that stopping states?

The referee raises three important points here: 1) one concerning statements that we made regarding state duration, 2) general issues related to state duration, 3) conclusions we draw based on the state durations.

With regards to the specific statements that >10% of the time is spent performing locomotor features >3 s: Figure 2—figure supplement 1 plots the fraction of transitions above a certain duration. To estimate how much time a fly spends in a state that is >a certain duration (t), we have to multiply N(t)*t, where N is the number of transitions of exactly duration t, and calculate P(>t) from the resulting quantity. We have now shown this calculation in Figure 2—figure supplement 2B. The number 10% comes from the mean curve (black line). When we examined the same distribution by state, we found (as implied by the reviewer above) that the stop state (State 2) dominates the long duration states. Interestingly, the fast states – states 9 and 10 and state 1 (meander) also has many long duration transitions. The medium speed states – states 4-7 rarely have long duration transitions. Therefore, the conclusion that “A fly moves at a relatively constant 𝑣̂|| and 𝑣̂⫠ for tens of step” is weaker than we originally thought. The general idea that flies can reside within the same state for many steps is true, and we have clarified the discussion in subsection “Odors affect behavior on a fine spatial scale” to reflect the same.

With regards to the general issue related to state durations – we present a new analysis which quantifies – 1) duration of HL states, 2) duration of low-level states, 3) No. of low-level state transition/transition into HL states. These analyses are detailed in Figure 2—figure supplement 2.

Finally we have clarified the conclusions we draw from state durations. We state clearly that the states of the HHMM are not necessarily the states employed by the system. In fact, as pointed out by another reviewer, there is no guarantee that there are discrete states at all. We completely agree with that sentiment and described explicitly in the discussion subsection “Limitations of the model”.

3) Subsection “Odors affect behavior on a fine spatial scale.”, paragraph two: Here you are creating a sequence of behavior based on averages and YET you claim that average data does not describe a fly's behavior but there are distinct strategies. Please clarify.

This is an excellent point. We have deleted the entire discussion.

Reviewer #2:

1) The model assumes that behavior is best modeled as set of discrete states (with the implication, I think, that this is how they are controlled neurally). The alternative possibility, alluded to briefly in subsection “Flies show considerable variability in locomotion despite employing the same locomotor features.”, is that some parts of the behavior could be better represented (or controlled) as continua. Since velocity is a continuous 1D variable, and since the fly must pass through intermediate velocities to transition from low to high velocity and vice-versa, I think this alternative should be considered. I am not suggesting the authors entirely revamp their model but I think this point could be discussed or considered a bit more prominently.

The most parsimonious way to think about the discrete vs. continuous is to think about the time-series of velocity and curvature as outputs of a dynamical system. In this framework, the HL states would represent the system being in a vicinity of a fixed point for the vast majority of time. We have included this idea in the discussion in subsection “Extracting observables from the trajectory”.

2) In the analysis of spatial structure (Figure 6), the role of time history should be considered/discussed. For example, behavior near the odor border might be different because the fly is more likely to have experienced no odor shortly before odor (or vice versa) than in the center of the arena. The responses of olfactory neurons are well known to show responses that depend on history over multiple time scales, so this point should be considered/discussed. Whether behavior could also be influenced by airflow at different parts of the chamber should be noted (without assuming the reader knows the details of the earlier paper). For example, could the airflow from the vacuum be causing the flies to slow down inside the odorized area, as in Yorozu et al., 2009?

The reviewer raises two issues here:

The first issue is that the behavior is not just spatially structured, but spatio-temporally structured. We agree with the reviewer. The spatial structure we present in Figure 6 is actually spatio-temporal structure. We have discussed this issue and added a figure (Figure 6—figure supplement 1) which shows that the behavior evolves as a function of time. However, this evolution is complete within the first 10 seconds or so. Unfortunately, because of 1) individuality among flies, 2) because we did not quantify the concentration of odor as a function of time (after the first minute), we cannot make a definitive analysis of the nature of the spatio-temporal structure in the data. The main point we are making in Figure 6 is that the HL states provide an important analytical tool for analyzing this structure.

The second issue relates to the airflow. We are quite confident that behavior in the arena is minimally impacted by airflow because the airflow we use is much smaller than almost any other study on fly olfaction, and is well-below the reported thresholds for anemotactic responses. More importantly, in the original study (Jung et al., 2015), we had performed controls in which we compared the effect of turning on the air, then the air and vacuum on the fly’s locomotion. We found that there was no observed effect of either the air or the air/vacuum. This issue is discussed in subsection “HHMM modeling”.

3) The findings about individuality are quite interesting. As mentioned above, the analysis in which the presence of odor can be predicted based on individual flies or clusters of flies, but not the whole dataset is quite interesting. However, as the authors note in the response to Reviewer 3, individuality in responses has also been investigated elsewhere. A typical analysis in these studies (de Bivort, Branson) is to compare behavior of the same individual at different time points and to show that they are more similar to themselves than to each other. The authors might consider splitting their data in time and repeating the analysis, although the time interval of the data here is relatively short (6 min). In addition, it would be nice to know if they could correlate their clusters with anything else about the flies. In the response to reviewers the authors allude to different behaviors produced by different genotypes. I think it would add to the interest of the study if these data were included.

There are two important points here. First, given our limited dataset, can we still assess whether the clusters are stable? This analysis is presented in Figure 7—figure supplement 4. The stability of the cluster depends on – 1) How different are the clusters from each other? If the clusters are very different, then only a small temporal sample is enough to assign flies to clusters. To assess how different the clusters are from each other, we assembled tracks of a given length from each fly by sampling data points at random. We found that there is a high probability that tracks >10 second in length were assigned to the original cluster. This analysis implies that the cluster means are sufficiently different that small chunks are representative.

2) How long of a time-sample does one need to describe the average behavior? We analyzed whether chunks of contiguous data points of a given length are representative of the original cluster. We performed two analyses: a) we asked whether chunks of a given length belong to the original cluster. We found that we could assign the flies to their original cluster at greater than chance level. b) We chose chunks either at the beginning or at the end of the track. Once again, we find that small chunks are predicted to be in their respective cluster at levels much greater than chance.

Second, we examined whether the clusters correlated with 1) time of recording, 2) starvation time, 3) age of the fly. We found no trends.

4) The comparison of HMM and HHMM models is nicely rigorous but seems highly technical for this journal. The authors should consider focusing the text on the conceptual conclusions that can be drawn from this comparison.

We agree with the reviewer. But, based on the previous set of reviews, we would like to keep that description. A large section of the audience has grave misunderstandings about HHMM vs. HMM. It is best to clarify these misconceptions at the outset.

Reviewer #3:

Before publication some work is needed to address questions about: 1) how to interpret HL behaviors (survival time statistics of HL states and relationship to the structure of a 10HL-5LL model); 2) error bars.Main comments:Error bars.I could not get a clear sense of how well the data is constraining the HHMM model. For example in Figure 5, what is the uncertainty of these distributions? Can the authors bootstrap the data? Likewise in Figure 6 how many data points go in each colored dot? What is the uncertainty of the probability plots as a function of radial distance? There are no error bars on these plots.

There are two sources of variability. The first corresponds to how well a model constrains the data. This source of variability can be quantified as follows: The standard deviation (SD) of the probability of HL states (P) in each scenario were calculated as follows:

SDj=Pj1-PjN-1

∀∀j∈1,2,3,…10HLstate

Where N is the total number of data points in each scenario. This standard deviation is extremely small and is shown in Figure 5.

The second source of variability is how different is the behavior of different flies in a given scenario. Getting to the bottom of this variability is one of the questions which drives the manuscript after Figure 6.

Time spent in HLs.One of the reasons HHMM seems to work better than HMM is that behaviors have extended durations and since the trajectory can jump around the LL states within a single HL state, that HL state can last long. Even though this is a key aspect of the model, there is no analysis or plot of the waiting time distribution in each HL state that would give an idea of the "duration" of such states during behavior and how that duration depends on the number of HL and LL states used in the HHMM model. The only thing I could find was Figure 2—figure supplement 1 (but this includes all states) and the statement made: "A fly moves at a relatively constant 𝑣̂|| and 𝑣̂⫠ for tens of step. This tendency means that a fly's locomotion can be decomposed into a small number of locomotor features – 10 features in the case of the model we present.". However, this main conclusion is provided without plots about it. Supporting this point with quantitative data analysis is important because the authors use it to interpret various aspect of the results. It is also important to know if some HLs state last much longer than other in order to interpret the data. This may also clarify why 10HLs each with 5LLs are used in the HHMM. See below.

This is an important point. We have now analyzed the duration distributions for both the HL and LL states (Figure 2—figure supplement 2). A short description of our findings is presented in the essential revisions and repeated below. We have analyzed the dwell times of each HL state (Figure 2—figure supplement 2). Specifically, we plot the cumulative distributions of the HL states. We also plot the percentage of time a fly spends in a state of a given duration. These plots show the differences in dwell time in different HL state.

The plot on dwell time – Figure 2—figure supplement 2 – also plots the average dwell time of the low-level states and the number of LL state transitions/transition into HL state as a function of the duration of the HL state. These data support the idea stated above: The number of transitions increase when the duration of HL states become longer. Thus, this analysis works exactly as the reviewers envision above, and as one would expect if the data has structure on multiple time scales.

Subsection " HHMM reveals that a fly’s locomotion is surprisingly structured in the velocity space".An interesting outcome underlined by the authors (last paragraph) is that the resulting HHMM accounts for 80% of all the data (within 85% confidence interval) using 10 locomotor features (10 HLs). Some of the HL states correspond to clearly distinct behavior, such as 1=meander, 2=stop-to-walk or 9=fast right turn, 10=fast left turn. However, other HLs seem to be part of a continuum, such as 4,5,7= medium speed walking, and it is not entirely clear why 3 distinct HLs are needed to describe medium speed walking. Some HLs seem to fit less well the data, e.g. HL 6. I could not find a clear explanation of why 10HLs each with 5LLs were chosen and whether fewer HLs would have worked as well. What would happen if the HHMM had less HL nodes but more LL nodes in each HL? A discussion of survival times in each HLs might help sort this out and provide a way to interpret states 4,5,7. Are these different behaviors or one behavior distributed over 3 HL?

The explanation of why we chose the model was in subsection “Model selection” of the Materials and methods. We have expanded this section to address the reviewer’s comments. To address the reviewer’s specific questions: The main reason for the choice of 10 HL and 5 LL states was *repeatability*.

We tried different combination of HL and LL states from 6-16 for HL states and 4-6 for LL states. There were some models with as few as 6 HL states that worked well. Models with 16 HL states rarely used more than 12 HL states. So, models with 6-12 HL states could work. However, multiple runs with 6 or 8 states gave widely different results, therefore we decided to use 10 states. With respect to the LL states, 4-6 states yielded similar results. If more than 6 LL states were chosen, many LL states remained unused.

We think that the range 6-12 is a fairly narrow range. One important point is that within this narrow range of states, the state duration in all the models we tried were very similar. The insensitivity of state duration made us confident that the structure we discovered is real.

A detailed analysis of the duration of the HL state seems consistent with the reviewer’s intuition that it would make more sense to combine those states 4-7 into a single state because these states have a shorter duration. Does reducing the number of HL state combine the three medium speed HL states into one state? The answer is largely a no (see Author response image 1). Although, the number of middle speed state (labeled in bold) decreases, even models with 8 HL state devote three separate states to medium speed locomotion. More importantly, the number of middle speed states stabilize to 4 states as the number of HL states is increased to 12.

One important limitation of our modelling approach (which we only realized in hindsight) is that it is difficult to compare across models because everything changes slightly– LL states change, HL states change. We are currently revising the model to keep the LL states fixed so that the modeling approach would mostly focus on how the HL select the LL states. This process will make it much easier to compare across models. Unfortunately, this change is non-trivial and beyond the scope of this manuscript.

**Author response image 1. respfig1:** The probability distribution of HL states from 3 models with different number of states – 10 state (top), model described in the manuscript), 12 states (middle), and 8 states (bottom). The number of middle speed states is the same between the 10-state model and the 12-state model, and decreases by 1 in the 8-state model.

Subsection “Locomotor features and implications for neural control of behavior”.

In the discussion the authors say that v_parallel lies within a narrow range that is distinct for each three state and that 3 states reflect a tight control on locomotion by the brain. They mention these locomotor state can persist up to 3 second/30 steps and suggest that this indicates that the brain controls locomotion in blocks of 10 steps. I find this conclusion drawn from the finding of these 3 clusters too speculative given the data: In Figure 3—figure supplement 1 and Figure 4 the distributions of v_par overlap significantly between HL 4,5 and 7. This is again related to the two points above. Finally, if HL 4 5 7 are really distinct behaviors then another possibility could be that they are needed to account for fly-to-fly variability in the data, and that for a single fly only one state is sufficient to describe medium speed walk.

Yes, we have changed the discussion around this claim.

[Editors' note: further revisions were requested prior to acceptance, as described below.]

Significant additional revisions are still required. The comments of the expert reviewers, below, are detailed and require full responses.Reviewer #2:[…] Two points need to be addressed before acceptance:1) How do the priors used in fitting constrain the structure of the transition matrices found for the models? In the Results section, the authors state that "the assumption of Markov dynamics with a sparse prior on state transitions penalizes the consideration of unlikely state transitions base upon recent history and future destinations." This suggests that a sparse prior was applied in fitting. However, later in the paper the authors state: "The transition probability matrix for the HMM was sparse, suggesting that from each state there are transitions to only a handful of other states." If a sparse prior was applied during fitting than the fact that the resulting transition matrix is sparse is not really a finding about the data.Similarly, the authors need to clarify what fitting priors or constraints were placed in the HHMM model. The Results state "This is because any HHMM- which puts very specific constraints on the transitions probability matrix- can be represented by an HMM but not vice-versa." What are the constraints on the transition probability matrix for the HHMM? The text says "The transition probability matrix is sparse- a vast majority of transitions from each HL state were to 2-3 other HL states." Was this imposed by fitting priors?

We have made the priors that we use more explicit (see subsection “Post-hoc analyses based on the HHMM model”). The description in subsection “Rationale for the choice of HHMM as the model and the model architecture” only refers to the fact that self-transitions are favored over transition to another state. The priors for all transitions from i^th^ – j^th^ state were the same.

With respect to HHMM versus HMM, this is already well-described in the manuscript. (“Rationale for the choice of HHMM as the model and the model architecture” and “Generation of synthetic flies”). Essentially, an HMM with 50 states allows transitions between each of the 50 states. HHMM with 10 HL states and 5 LL states only allows a fraction of these transitions because only LL states within each HL state can transition amongst each other. This property also means that HHMM can always be represented by HMM and not vice-versa.

2) Parts of the manuscript are somewhat long and repetitive. I think the manuscript could be productively shortened to have greater impact on its readers.For example, in the section titled "HHMM reveals that a fly's locomotion is surprisingly structured in the velocity space" the last paragraph is devoted to restating the major conclusion of the first section: HHMM performs better than HMM. I think this could be reduced or folded into section one, which focusses on this comparison, and the section focused more exclusively on describing the HL states uncovered by the model.In the Discussion subsection “Flies show considerable variability in locomotion despite employing the same locomotor features” paragraphs two and three are highly descriptive of the individual fly data and somewhat repetitive with parts of the Results subsection “Both locomotion and an odor’s effect on locomotion is fly dependent” (already rather long). I think these sections could be condensed to make the major points about individual variability in the Results, and use the Discussion mostly to compare these findings with the results of other studies.

We agree with the reviewer. We have not only made the two specific changes suggested above. We have also taken a careful look at the manuscript, and by cutting repeated segments and tightening the writing, we have reduced the word count (except for the Materials and methods) by ~2000 words.

Reviewer #3:

The authors have addressed most of my concerns expect the important one regarding the confidence interval on the distributions in Figure 5. What I am asking is for the authors to use bootstrapping to extract many sample distributions from subsets of the data in order to get a confidence interval on how these bins are populated by the data and how significant the changes are between before and during odor exposure.

Apologies for not fully understanding your query. We have performed the bootstrapping. Confidence intervals and significance are noted in Figure 5.